



# Quantifying and attributing time step sensitivities in present-day climate simulations conducted with EAMv1

Hui Wan[1], Shixuan Zhang[1], Philip J. Rasch[1], Vincent E. Larson[2,1], Xubin Zeng[3], and Huiping Yan[4,1]

[1]Atmospheric Sciences and Global Change Division, Pacific Northwest National Laboratory
[2]Department of Mathematical Sciences, University of Wisconsin – Milwaukee
[3]Department of Hydrology and Atmospheric Sciences, University of Arizona
[4]School of Atmospheric Science, Nanjing University of Information Science and Technology

**Correspondence:** Hui Wan (Hui.Wan@pnnl.gov)

**Abstract.** This study assesses the relative importance of time integration error in present-day climate simulations conducted with the atmosphere component of the Energy Exascale Earth System Model version 1 (EAMv1) at 1-degree horizontal resolution. We show that a factor-of-6 reduction of time step size in all major parts of the model leads to significant changes in the long-term mean climate. Examples of such changes include warming in the lower troposphere, cooling in the tropical

and subtropical upper troposphere, as well as decreases of relative humidity throughout the troposphere accompanied by cloud fraction decreases. These changes imply that the reduction of temporal truncation errors leads to a notable although unsurprising degradation of agreement between the simulated and observed present-day climate; the model would require retuning to regain optimal climate fidelity in the absence of those truncation errors.

A coarse-grained attribution of the time step sensitivities is carried out by separately shortening time steps used in various

components of EAM or by revising the numerical coupling between some processes. Our analysis leads to the counter-intuitive finding that the marked decreases in the subtropical low-cloud fraction and total cloud radiative effect are caused not by the step size used for the collectively subcycled turbulence, shallow convection and stratiform cloud macro- and microphysics parameterizations but by the step sizes used outside the subcycles. Further analysis suggests that the coupling frequency between the subcycles and the rest of EAM has a substantial impact on the marine stratocumulus decks while the deep convection

parameterization has a significant impact on trade cumulus. The step size of the cloud macro- and microphysics subcycles appears to have a primary impact on cloud fraction at most latitudes in the upper troposphere as well as in the mid-latitude near-surface layers. Impacts of step sizes used by the dynamical core and radiation appear to be relatively small. These results provide useful clues to help better understand the root causes of time step sensitivities in EAM. The experimentation strategy used here can also provide a pathway for other models to identify and reduce time integration errors.

## 1  Introduction

Atmospheric General Circulation Models (AGCMs) simulate physical and chemical processes in the Earth's atmosphere by solving a complex set of ordinary and partial differential equations. It is highly desirable that the numerical methods used for solving those equations produce relatively small errors, so that the behavior of an AGCM reflects the inherent characteristics





of the continuous model formulation that describes the model developers' understanding of the underlying physical and chem-
ical processes (see, e.g. Beljaars, 1991; Beljaars et al., 2018). However, various studies have shown examples where temporal
discretization methods in AGCMs, especially those used in the parameterization of unresolved processes or in the coupling
between processes, produce large errors that significantly affect key features of the numerical results (Wan et al., 2013; Get-
telman et al., 2015; Beljaars et al., 2017; Donahue and Caldwell, 2018; Zhang et al., 2018; Barrett et al., 2019). These results
are not surprising, given the relatively short time scales associated with parameterized processes such as clouds and turbulence
and the relatively long time steps (typically on the order of tens of minutes) used by current global atmospheric GCMs.

This study attempts to take a first step towards assessing and addressing time integration issues associated with physics
parameterizations in the atmospheric component of the U.S. Department of Energy's Energy Exascale Earth System Model
version 1, hereafter referred to as EAMv1 (Rasch et al., 2019). The study contains two parts:

- The first part is an initial assessment of the relative importance of time integration errors in present-day climate simula-
tions conducted with EAMv1. This is done by using an intuitive and practical metric, namely the magnitude of changes
in the model's long-term climate resulting from a substantial (in our case a factor of 6) reduction of the time step sizes
used in all major components of the model (e.g., resolved dynamics, parameterized radiation, stratiform clouds, deep
convection, and the numerical coupling of different processes). As we show in Section 3, the consequent changes in
EAMv1's 10-year climate statistics lead to a notable and unsurprising degradation in agreement between the simulations
and observations because errors which were previously compensated for using parameter tuning are no longer present in
the model solutions, and no retuning was performed in this study.

- In the second part, a series of sensitivity experiments are conducted and analyzed to identify which components of EAM
are responsible for the changes in cloud fraction and cloud radiative effects. The purpose is to provide clues for future
studies that investigate the root causes of the sensitivities to model time step.

The rest of this paper proceeds as follows. Section 2 provides an overview of the EAM model, introduces the time step sizes
used by its main components, and briefly describes the numerical methods used for process coupling. The common setup of the
present-day climate simulations and the methods used for assessing statistical significance of the sensitivities are also described.
Section 3 presents the impact of a proportional, factor-of-6 step size reduction in all major components of EAMv1. Section 4
presents results from additional numerical experiments to attribute the sensitivities in cloud fraction and cloud radiative effects
shown in Section 3. The conclusions are drawn in Section 5.

## 2  Model and simulation overview

### 2.1  EAMv1

EAMv1 is a global hydrostatic AGCM. The dynamical core solves the so-called primitive equations using a continuous
Galerkin spectral-element method for horizontal discretization on a cubed-sphere mesh (Dennis et al., 2012; Taylor et al.,





2009). The vertical discretization uses a semi-Lagrangian approach in a pressure-based terrain-following coordinate (Lin, 2004). Main components of the parameterizations suite include the solar and terrestrial radiation (Mlawer et al., 1997; Iacono et al., 2008), deep convection (Zhang and McFarlane, 1995; Richter and Rasch, 2008; Neale et al., 2008), turbulence and shallow convection (Golaz et al., 2002; Larson et al., 2002; Larson and Golaz, 2005; Bogenschutz et al., 2013), stratiform cloud microphysics (Morrison and Gettelman, 2008a; Gettelman and Morrison, 2015; Gettelman et al., 2015; Wang et al., 2014),

aerosol lifecycle and aerosol-cloud interactions (Liu et al., 2016; Wang et al., 2020), and land surface processes (Oleson et al., 2013).

The so-called low-resolution (or standard) configuration of EAM uses a horizontal grid-spacing of approximately 100 km. The vertical grid consists of 72 layers covering an altitude range from the Earth's surface to 0.1 hPa (64 km), with layer thicknesses ranging from 20–100 m near the surface to about 600 m in the free troposphere up to the lower stratosphere. This

1-degree configuration is used as one of the workhorses both for model development and in multi-decade simulations targeted at scientific investigations. A more detailed description of EAMv1 can be found in Rasch et al. (2019) and Xie et al. (2018).

Various different time integration methods and time step sizes are used by different parts (hereafter referred to as components) of EAMv1. These are mostly explicit or implicit methods using fixed step sizes. For example, in the dynamical core, the temperature, horizontal winds, and surface pressure equations are advanced in time using an explicit five-stage third-order

Runge-Kutta method (Kinnmark and Gray, 1984; Guerra and Ullrich, 2016; Lauritzen et al., 2018). The horizontal tracer advection uses a three-stage second-order strong-stability-preserving Runge-Kutta method (Spiteri and Ruuth, 2002; Guba et al., 2014; Lauritzen et al., 2018). Some parameterizations, e.g., sedimentation of rain and snow and turbulent mixing of aerosols, are subcycled using dynamically determined step sizes.

The primary method used for process coupling isolates a model component representing an atmospheric process or a set of

processes within EAM by updating model state variables (e.g., winds, temperature, pressure, and tracer concentrations) that may have been updated by a preceding component. This method is referred to as "time splitting" in Williamson (2002) and in Lauritzen et al. (2018), and is given the name "sequential update splitting" in Donahue and Caldwell (2018). In wider numerical modeling communities, the method is often referred to as "operator splitting" (e.g., Sportisse, 2000). We denote the step size of this splitting/coupling by $\Delta t_{\mathrm{CPLmain}}$. For completeness, it is worth noting that in some parts of EAM, tendencies predicted

by one atmospheric process are also passed on to a subsequently calculated process. An example like this can be found in Figure 2b of Zhang et al. (2018) which depicts the coupling between the parameterized physics and the resolved dynamics. These instances of a different process coupling method in the default EAMv1 are not investigated in this study, and hence are not further detailed here. The sequence of calculations used in EAMv1 is schematically depicted in Figure 1a. The following step sizes are used as defaults in the 1-degree configuration of EAMv1:

– $\Delta t_{\mathrm{CPLmain}}$ = 30 min.

– Within the resolved dynamics, the vertical discretization (remapping) takes step sizes of $\Delta t_{\mathrm{remap}}$= 15 min, each of which is further divided into 3 substeps of $\Delta t_{\mathrm{adv}}$= 5 min for the horizontal advection of temperature, momentum, and tracers.





**Figure 1.** (a) A flowchart showing the sequence of calculations in EAMv1. (b) Time step sizes used in the default EAMv1 for 100 km horizontal resolution (simulation v1_CTRL in this paper). Different colors in panel (b) indicate step sizes controled by different namelist variables in the model source code. Step sizes shown in the same color have the same value. More details can be found in Section 2.1.





These step sizes can be changed independently as long as $\Delta t_{\mathrm{remap}}$ is a multiple of $\Delta t_{\mathrm{adv}}$ and $\Delta t_{\mathrm{CPLmain}}$ is a multiple of $\Delta t_{\mathrm{remap}}$.

– Deep convection uses $\Delta t_{\mathrm{deepCu}}$= 30 min; this tied to (i.e., has to be the same as) $\Delta t_{\mathrm{CPLmain}}$.

     – The parameterizations of stratiform and shallow cumulus clouds include two elements: (1) a treatment of turbulence and shallow convection named Cloud Layers Unified By Binormals (CLUBB, Golaz et al., 2002; Larson et al., 2002; Larson and Golaz, 2005), which we refer to for brevity as cloud macrophysics in this paper, and (2) a treatment for aerosol activation (formation of cloud liquid and ice particles) and further evolution of cloud condensate which we refer

95       to as cloud microphysics. These two elements are subcycled together using time steps of $\Delta t_{\mathrm{macmic}}$= 5 min following Gettelman et al. (2015). To facilitate discussions later in this paper, we use the notation $\Delta t_{\mathrm{CPLmacmic}}$ to denote the step size used for coupling the collectively subcycled cloud macro-/microphysics with the rest of EAM. The default EAMv1 has $\Delta t_{\mathrm{CPLmacmic}} \equiv \Delta t_{\mathrm{CPLmain}}$ but an alternative is discussed in Section 4.3. CLUBB can be further subcycled with respect to $\Delta t_{\mathrm{macmic}}$, but that is not done in either the default EAMv1 or in any of the simulations presented in this paper.

– Heating/cooling rates resulting from shortwave (SW) and longwave (LW) radiation are calculated every hour, i.e., $\Delta t_{\mathrm{rad}}$= 60 min. This means radiation is supercycled with respect to all the other parameterizations as well as the resolved dynamics. During every other time step of $\Delta t_{\mathrm{CPLmain}}$= 30 min when the radiation parameterization is not exercised, the tendencies saved from the previous 30 min time step are used to update the atmospheric state.

     – Miscellaneous other atmospheric processes, e.g., gravity wave drag and the sedimentation, dry deposition, and micro-

105       physics of aerosols, are coupled with each other and with the processes listed above at time intervals tied to $\Delta t_{\mathrm{CPLmain}}$. The coupling to land surface happens at intervals of $\Delta t_{\mathrm{CPLmain}}$ by default; it can be changed to longer multiples of $\Delta t_{\mathrm{CPLmain}}$ but that again is not explored in this study.

These various step sizes are schematically depicted in Figure 1b. Their relationship in the default EAMv1 can be summarized as follows:

$$\Delta t_{\mathrm{CPLmain}} \quad = \quad 2\Delta t_{\mathrm{remap}} = 6\Delta t_{\mathrm{adv}} \tag{1}$$

$$\Delta t_{\mathrm{CPLmain}} \quad \equiv \quad \Delta t_{\mathrm{CPLmacmic}} = 6\Delta t_{\mathrm{macmic}} \tag{2}$$

$$\Delta t_{\mathrm{CPLmain}} \quad \equiv \quad \Delta t_{\mathrm{deepCu}} \tag{3}$$

$$\Delta t_{\mathrm{CPLmain}} \quad = \quad 0.5\Delta t_{\mathrm{rad}} . \tag{4}$$

The equivalent sign ($\equiv$) indicates step sizes that are tied together in the default EAMv1.

## 2.2 EAMv0

To provide context and serve as a reference for the evaluation of time step sensitivity in EAMv1, we also present one simulation using EAMv0, i.e., EAMv1's most recent predecessor. EAMv0 uses the same dynamical core and large-scale transport





algorithms as in v1, but the vertical grid has only 30 layers. Many of the parameterizations differ from EAMv1. The parameterization of turbulence and shallow convection follows Park and Bretherton (2009), the cloud macrophysics parameterization

follows Park et al. (2014), and the cloud microphysics parameterization is described in Morrison and Gettelman (2008b). The time integration methods and step sizes are very similar to those in EAMv1, except that the cloud macro- and microphysics parameterizations are not subcycled (i.e., they use a 30 min step size).

### 2.3 Present-day climate simulations

A series of 10-year simulations were conducted using an experimental setup commonly exercised in the development and eval-

uation of EAM and its predecessors. The model was configured to simulate recent climatological conditions by selecting values of the Earth's orbital conditions, aerosol emissions and greenhouse gas concentrations, land use, and sea surface temperatures and sea ice coverage characteristics of the recent past (around year 2000). The sea surface temperature and sea ice cover were prescribed using monthly climatological values that repeated each year. Prognostic equations were used to produce evolving descriptions of the atmosphere and land states. The simulations used initial conditions written out by a previously performed

multiyear simulation. Some of the model configurations used in our sensitivity experiments produced climate statistics that differed substantially from the default configuration; therefore, to avoid characterizing the initial adjustment phase, a 4 month spin-up was performed and neglected in each simulation, and the following 10 years were analyzed.

Simulations were first conducted with EAMv0 or v1 using their default time step sizes. These are labelled "v0_CTRL" and "v1_CTRL", respectively. In a second v1 simulation called "v1_All_Shorter", the various step sizes listed in Eqs. (1)–(4) were

proportionally reduced by a factor of 6, cf. Table 1 and flowchart in Figure A1a. This reduction gives a step size of 5 min for most of the parameterizations and for process coupling, which is significantly shorter than the default, but is still practically affordable for sensitivity studies. Results from these three simulations are discussed in Section 3. Additional simulations were also conducted with the v1 model to allow the differences between v1_All_Shorter and v1_CTRL to be attributed to specific sets of processes and time stepping algorithms. The experimental design is summarized in Tables 1 and A1, groups II and III.

The attribution process is summarized in Figure 2 with the detailed results discussed in Section 4.

### 2.4 Statistical tests

The analyses presented in this paper focus primarily on 10-year mean annual averages. To distinguish signals of time step sensitivity from noise caused by natural variability, the two-sample $t$-test was applied to pairs of simulations conducted with different step sizes, with the test statistic constructed using annual averages. A significance level of 0.05 was chosen to de-

termine whether differences between a pair of 10-year averages were statistically significant. This method of two-sample $t$-test has been used in the diagnostics package of the National Center for Atmospheric Research (NCAR) Atmosphere Model Wording Group (AMWG) who developed predecessors of EAM (http://www.cesm.ucar.edu/working_groups/Atmosphere/ amwg-diagnostics-package/) .

Considering that the sample size of 10 is relatively small, we also conducted statistical testing using monthly mean model

output. Serial correlation in monthly averages was addressed by using the paired $t$-test and the effective sample size (Zwiers





**Table 1.** List of climate simulations conducted in this study. The numbers given in the main part of the table are the ratio of each step size (or $\Delta t_{\mathrm{DeepCu}}/\tau$) relative to its value in the default model running at 1-degree horizontal resolution. The meaning and default values of the various $\Delta t$ values are explained in Section 2. Here $\tau$ refers to the prescribed (fixed) time scale in the deep convection parameterization for releasing the convective available potential energy (CAPE), the default value of which is 3600 s. The namelist variables in EAM used to configure these simulations are listed in Table A1. Flowcharts of the EAMv1 simulations are shown in Figures 1, 12, A1, and A2.

| Group | Simulation | Description | Flowchart | Ratio of time step size relative to default | | | | | | | Ratio of $\Delta t_{\mathrm{deepCu}}/\tau$ relative to default |
| --- | --- | --- | --- | --- | --- | --- | --- | --- | --- | --- | --- |
| | | | | $\Delta t_{\mathrm{remap}}$ | $\Delta t_{\mathrm{adv}}$ | $\Delta t_{\mathrm{CPLmain}}$ | $\Delta t_{\mathrm{deepCu}}$ | $\Delta t_{\mathrm{CPLmacmic}}$ | $\Delta t_{\mathrm{macmic}}$ | $\Delta t_{\mathrm{rad}}$ | |
| 0 | v0_CTRL | Sect. 2.2 | - | 1 | 1 | 1 | 1 | 1 | 1 | 1 | 1 |
| I | v1_CTRL | Sect. 2.1 | Fig. 1 | 1 | 1 | 1 | 1 | 1 | 1 | 1 | 1 |
| I | v1_All_Shorter | Sect. 2.3 | Fig. A1a | 1/6 | 1/6 | 1/6 | 1/6 | 1/6 | 1/6 | 1/6 | 1/6 |
| II | v1_MacMic_Shorter | Sect. 4.1 | Fig. A1b | 1 | 1 | 1 | 1 | 1 | 1/6 | 1 | 1 |
| II | v1_All_Except_MacMic_Shorter | Sect. 4.1 | Fig. A2a | 1/6 | 1/6 | 1/6 | 1/6 | 1/6 | 1 | 1/6 | 1/6 |
| III | v1_CPL+DeepCu_Shorter | Sect. 4.3.1 | Fig. A2b | 1/3 | 1 | 1/6 | 1/6 | 1/6 | 1 | 1 | 1/6 |
| III | v1_CPL+DeepCu+Tau_Shorter | Sect. 4.3.3 | Fig. A2b | 1/3 | 1 | 1/6 | 1/6 | 1/6 | 1 | 1 | 1 |
| III | v1_Dribble | Sect. 4.3.2 | Fig. 12 | 1 | 1 | 1 | 1 | 1/6 | 1 | 1 | 1 |





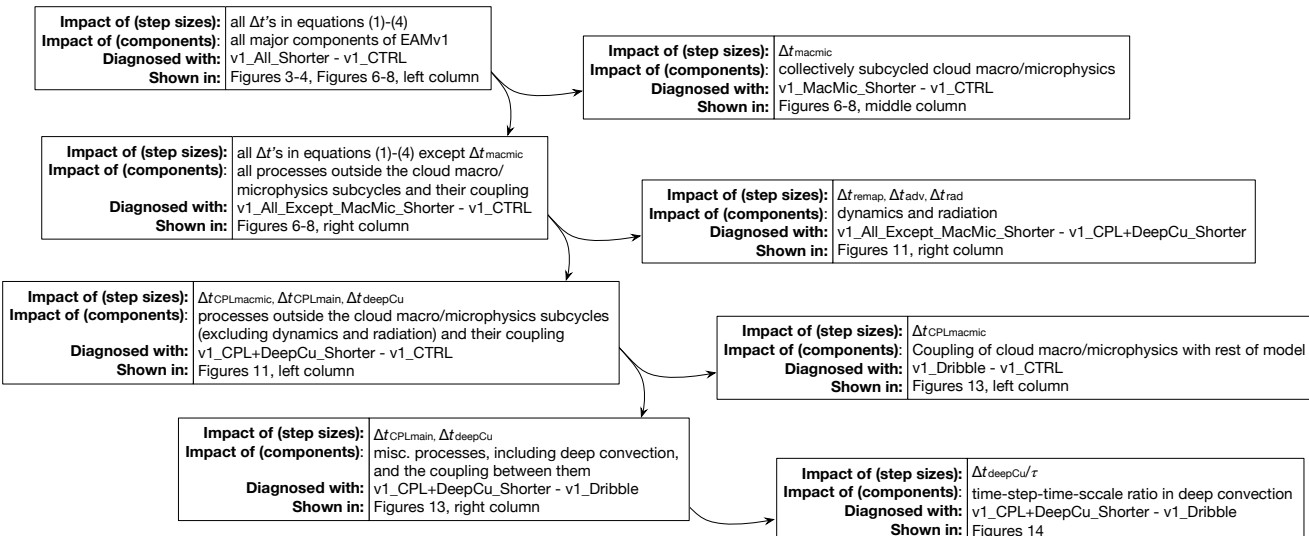

**Figure 2.** Schematic showing the attribution of time step sensitivities using simulations listed in Tables 1 and A1. Flowcharts of the simulations are shown in Figures 1, 12, A1, and A2.

and von Storch, 1995). For example, to assess the significance of the differences between simulations A and B at a certain geographical location, we used the time series of monthly mean $A - B$ (which had 120 data points in the monthly time series) to construct the test statistic for a one-sample $t$-test, taking into account the autocorrelation in the time series. A significance level of 0.05 was chosen to determine whether the mean of the differences was statistically zero.

We processed all the difference plots shown in the paper using both methods. The two methods turned out to give rather consistent results overall. They disagree only at a small portion of grid points associated with relatively small climate differences. The key signatures of time step sensitivity discussed below were considered statistically significant by both methods. We chose to show results from the two-sample test in the paper to be consistent with the AMWG diagnostics package.

## 3   Impact of proportional step size reductions in all major processes

The first question we attempt to answer is whether the characteristics of EAMv1's present-day climate are substantially affected by the choices of time step sizes. This is done by comparing simulations v1_CTRL and v1_All_Shorter. To put the magnitude of the differences into context, we also show some representative results from v0_CTRL.

### 3.1   Time step sensitivities in EAMv1

It turns out that the proportional factor-of-6 step size reduction in all major components of the v1 model leads to systematic
changes in the simulated long-term climate. In the middle column of Figure 3, the differences in 10-year-mean zonal averages

**Figure 3.** Left column: 10-year mean, zonally averaged air temperature (T), specific humidity (Q), relative humidity (RH), and cloud fraction (f) in simulation v1_CTRL. Middle column: differences between v1_All_Shorter and v1_CTRL. Right column: relative differences with respect to v1_CTRL. Statistically insignificant differences are masked out in white. The simulation setups are described in Section 2.3, Table 1 and Table A1, group I. The flowcharts can be found in Figures 1 and A1a.



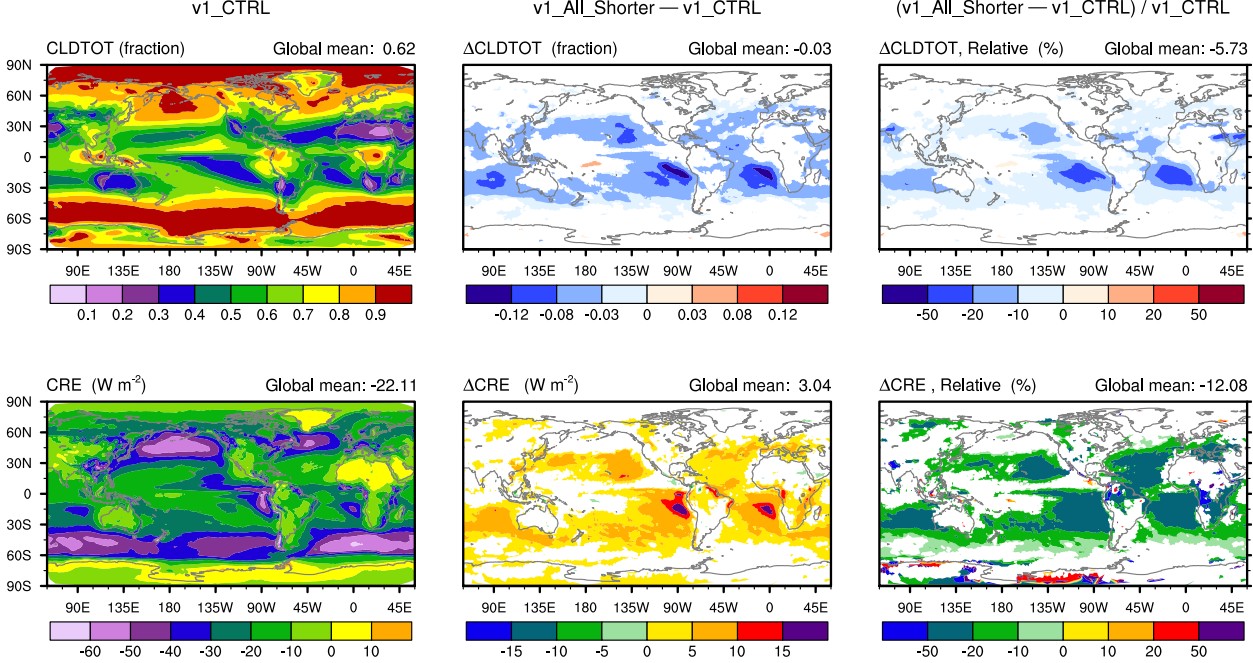

**Figure 4.** Left column: 10-year mean geographical distribution of total cloud cover (CLDTOT, upper row) and total cloud radiative effect (CRE, lower row) in v1_CTRL. Middle column: differences between v1_All_Shorter and v1_CTRL. Right column: relative differences with respect to v1_CTRL. Statistically insignificant differences are masked out in white. The simulation setups are described in Section 2.3, Table 1, and Table A1, group I. The flowcharts can be found in Figures 1 and A1a.

between v1_All_Shorter and v1_CTRL are shown for air temperature (T), specific humidity (Q), relative humidity (RH) and cloud fraction (f). The relative differences normalized by the corresponding values in v1_CTRL are shown in the right column. (Relative differences in T are not a useful measure and hence not included.) Statistically *insignificant* differences are masked out in white. The figure reveals that the step size reduction leads to warming of up to 0.5 K in the subtropical and mid-latitude

near-surface layers and cooling of similar magnitudes in the tropical mid- and upper-troposphere (Figure 3, second panel in first row). In the middle and low latitudes, the air dries at most altitudes in the troposphere, showing typical decreases of 1% to 10% in both specific and relative humidity (Figure 3, right column). Cloud fraction also decreases (Figure 3, bottom row). The largest changes appear in three regions: in the upper troposphere where ice clouds dominate, in the subtropical lower troposphere where stratocumulus and trade cumulus prevail, and in the mid-latitude near-surface layers.

The 10-year mean geographical distribution of total cloud cover and total cloud radiative effect (CRE) are shown in Figure 4. Here the signatures of time step sensitivity appear to be dominated by changes in subtropical stratocumulus and trade cumulus clouds over the oceans. The largest local changes are on the order of -10% to -50% for cloud cover and -20% to -50% for CRE. The global mean CRE weakens by about 3 W m$^{-2}$, corresponding to a relative change of -12%.





## 3.2 Comparison with observations and EAMv0

A recent evaluation of EAMv1 has shown that the simulated present-day climate is cooler and drier than reanalysis in the
tropical upper troposphere while the CRE in the major marine stratocumulus regions are weaker compared to satellite products
(cf. Figures 3, 4, and 10 in Rasch et al., 2019). Comparing those results with the time step sensitivities shown in Figures 3 and
4, one gets the impression that model biases in v1_All_Shorter are likely to be larger than those in v1_CTRL. Here a model bias
is defined as a deviation from the real-world observation. To obtain a comprehensive and yet concise assessment of the impact
of time step sizes on model fidelity, we follow the spirit of Donahue and Caldwell (2018) and use the collection of reanalyses
and satellite products listed in Table 5 to evaluate the fidelity of v1_CTRL and v1_All_Shorter. The results are presented in
Figure 5, where the upper panel shows the relative errors in the simulated global averages and the lower panel shows the relative
errors in global patterns. The relative pattern error is defined as the centered root-mean-square (RMS) difference between the
simulated and observed patterns, normalized by the RMS of the observed pattern. A "pattern" here refers to the annual mean,
global, geographical distribution of a physical quantity. The model results used in the calculations were 10-year averages. The
observational data was averaged over the years indicated in Table 5. The biases in v0_CTRL are also included in the figure for
comparison.

Figure 5 reveals that model biases in both the global mean (upper panel) and the spatial pattern (lower panel) are larger in
v1_All_Shorter for most of the physical quantities examined here; the magnitude of the differences is comparable to the differ-
ences between v1_CTRL and v0_CTRL. For clarification, we note that v1_CTRL and v0_CTRL have different characteristic
biases due to the substantial changes in the parameterizations and the vertical resolution. For example, Figure A3 shows that
the shortwave CRE biases in the low latitudes are dominated by overestimation in the monsoon regions in v0 and underesti-
mation associated with the marine stratocumulus decks and over the warm pool. If we were to compare the local differences
between v1_All_Shorter and v1_CTRL with the local differences between v0_CTRL and v1_CTRL, then the time step caused
differences will appear to be substantially smaller than the differences caused by changes in parameterizations and vertical
resolution, as should be expected. On the other hand, when comparing all three simulations (v1_All_Shorter, v1_CTRL, and
v0_CTRL) with observations, we see that the degradation of model fidelity caused by reducing step sizes in v1 has a magnitude
similar to the fidelity differences between v1_CTRL and v0_CTRL.

Given that substantial efforts have been made to tune the default EAMv1, i.e., to adjust the values of uncertain parameters
in the model's equations in order to improve the match between the simulations and observations (see, e.g., Xie et al., 2018;
Rasch et al., 2019), a degradation of model fidelity associated with shortened time steps is not surprising. Assuming the
time integration methods used in EAMv1 are mathematically consistent and convergent, one would expect shorter time steps
to give *numerically* more accurate results. The results shown in Figure 5 indicate that the default EAMv1 contains sizable
time integration errors that are compensated by parameter tuning or by other sources of model error. While the existence of
compensating errors is undesirable, it seems to be the widely recognized and accepted status quo. Reducing time integration
errors would sacrifice the immediate results and temporarily degrade model fidelity, but it would also provide the opportunity to
first expose and then address errors from other sources, hence eventually leading to a model that gives correct results for correct





**Table 2.** List of observational data and EAM's output used for evaluating the model biases. The observational data were obtained from NCAR AMWG diagnostics package (http://www.cgd.ucar.edu/amp/amwg/diagnostics/plotType.html).

| Physical quantity | Source of observation | EAM output |
|---|---|---|
| Surface longwave downwelling flux | ISCCP (1983–2000) | FLDS |
| Surface net longwave flux | ISCCP (1983–2000) | FLNS |
| TOA upward longwave flux | CERES-EBAF (2000–2010) | FLUT |
| TOA clearsky upward longwave flux | CERES-EBAF (2000–2010) | FLUTC |
| TOA longwave cloud forcing | CERES-EBAF (2000–2010) | LWCF |
| Surface net shortwave flux | ISCCP (1983–2000) | FSNS |
| TOA net shortwave flux | CERES-EBAF (2000–2010) | FSNTOA |
| TOA clearsky net shortwave flux | CERES-EBAF (2000–2010) | FSNTOAC |
| Shortwave cloud radiative effect | CERES-EBAF (2000–2010) | SWCF |
| Total cloud amount | CloudSat (2007–2010) | CLDTOT |
| 200 hPa zonal wind | JRA25 (1979–2004) | U |
| 500 hPa geopotential height | JRA25 (1979–2004) | Z3 |
| Precipitation rate | GPCP (1979–2009) | PRECT |
| Total precipitable water | NVAP (1988–1999) | TMQ |
| Sea level pressure | ERAI (1989–2005) | PSL |
| Surface latent heat flux | JRA25 (1979–2004) | LHFLX |
| Surface sensible heat flux | JRA25 (1979–2004) | SHFLX |
| Surface stress | ERS (1992–2000) | TAUX, TAUY |
| 2m air temperature | LEGATES (1920–1980) | TREFHT |
| Sea level temperature on land | NCEP (1979–1998) | TS |

reasons. As a first step towards reducing time-stepping errors in EAMv1, the next section identifies the model components that have caused the differences between v1_All_Shorter and v1_CTRL. While a number of physical quantities are shown in

Figure 5, the analysis in the remainder of the paper focuses on cloud fraction and CRE. Extension of the analysis to additional variables, such as temperature, humidity, precipitation, and winds, are left to future studies.

## 4    Attributing time step sensitivities in cloud fraction and CRE

The primary method used here for attributing the differences between v1_All_Shorter and v1_CTRL is to carry out sensitivity experiments in which we vary the step sizes used by different subsets of EAM's components. These experiments are sum-

marized in Tables 1 and A1, groups II and III, and are described in detail in the following subsections. An overview of the attribution process is provided in Figure 2 with pointers to the figures.





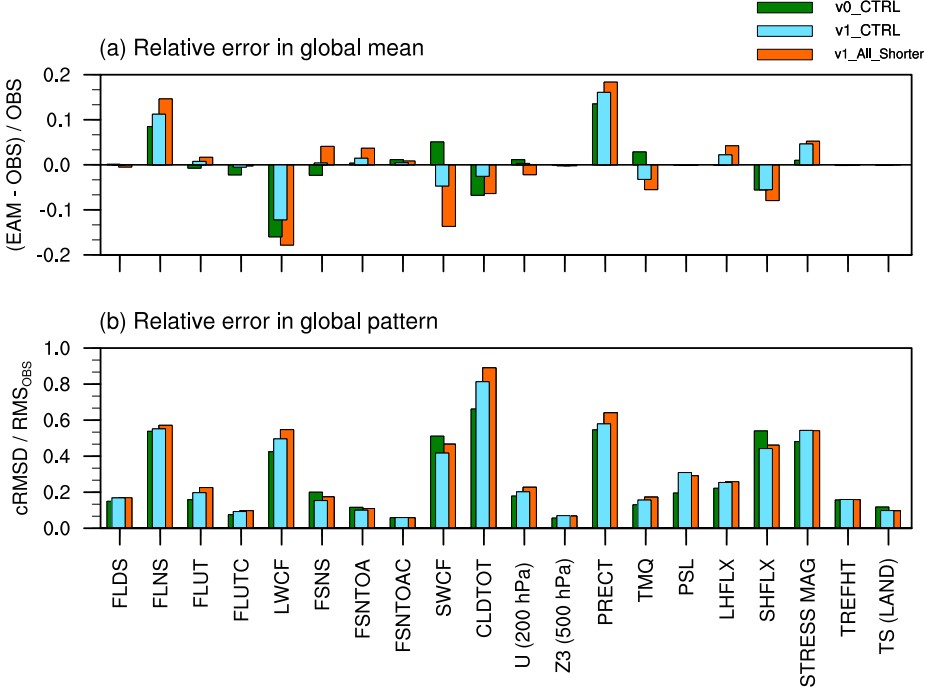

**Figure 5.** Comparison of 10-year-mean climate simulated by v1_All_Shorter, v1_CTRL and v0_CTRL with various reanalyses and satellite products. The upper panel shows relative errors in the simulated global averages. The lower panel shows the relative error in the simulated geographical distributions, as measured by the centered root-mean-square differences (cRMSD) between model results and the observations normalized by the root-mean-square of the observed global distribution (RMS$_{\text{OBS}}$). The long names of the physical quantities labeled along the x-axis are listed in Table 2 together with the sources of observational data.

## 4.1 Stratiform cloud parameterizations versus the rest of EAMv1

The key signatures in the total cloud cover and CRE changes are seen in the subtropics where marine stratocumulus and trade cumulus are the dominant cloud types (Figure 4). Since these clouds are strongly affected by turbulence, shallow convec-
tion, and microphysics, it seems natural to link those changes with the corresponding parameterizations. Two hypotheses are explored here:

- Hypothesis 1: The differences in total cloud cover and CRE seen in the subtropics between v1_All_Shorter and v1_CTRL are caused by time integration errors in the stratiform and shallow cumulus cloud macro- and microphysics parameter- izations, i.e., CLUBB, aerosol activation, and MG2. Turbulence and cloud microphysics are known to have relatively
short characteristic time scales. The 5 min step size ($\Delta t_{\text{macmic}}$= 5 min) used in the default EAMv1 cannot sufficiently resolve those short time scales and hence gives numerically inaccurate results.



– Hypothesis 2: The differences in total cloud cover and CRE seen in the subtropics between v1_All_Shorter and v1_CTRL are caused by time integration errors in parts of EAM other than the cloud macro- and microphysics parameterizations or in process coupling. In v1_All_Shorter, the reduction of time integration error in those other components or their coupling with cloud macro- and microphysics results in a different atmospheric environment being provided to CLUBB, and hence triggering different responses of the shallow cumulus and stratiform clouds.

The two sensitivity experiments listed in group II of Tables 1 and A1 were carried out to investigate these hypotheses: Simulation **v1_MacMic_Shorter** (see also flowchart in Figures A1b) sets the step size of the collectively subcycled shallow cumulus and stratiform cloud parameterizations, i.e., CLUBB, aerosol activation, and MG2, to 1/6 of the default value, i..e, $\Delta t_{\mathrm{macmic}}$ = 5/6 min as in v1_All_Shorter. The rest of EAMv1 used the identical time integration strategy (and thus equivalent time step sizes) as in v1_CTRL. In other words, within each of the main coupling time step $\Delta t_{\mathrm{CPLmain}}$ = 30 min, instead of 6 invocations of the cloud macro- and microphysics parameterizations with 5 min time steps, there were 36 invocations with 5/6 min time steps. The differences between results from v1_MacMic_Shorter and v1_CTRL are attributed to differences in $\Delta t_{\mathrm{macmic}}$ which controls the step sizes used in CLUBB and MG2 as well as the interactions between the processes within each subcycle.

Simulation **v1_All_Except_MacMic_Shorter** (see also flowchart in Figure A2a) has the opposite setup, i.e., $\Delta t_{\mathrm{macmic}}$= 5 min as in v1_CTRL while the rest of EAMv1 used the much shorter steps as in v1_All_Shorter. The differences in model climate between v1_All_Except_MacMic_Shorter and v1_CTRL are attributed to reduced step sizes for all model components outside the cloud macro- and microphysics subcycles and for the coupling (i.e., information exchange) between the subcycles and the other components (cf. Eqs. (1)–(4)).

The 10-year mean difference plots shown in Figures 6–8 indicate that the changes in long-term climate caused by $\Delta t_{\mathrm{macmic}}$ and other step sizes are both non-negligible but show different signatures. The zonal mean temperature, humidity, and cloud fraction differences shown Figure 6 reveal that:

– The warming as well as decreases in cloud fraction *around 850 hPa in the subtropics* (Figure 6a and j) is primarily attributable to shorter step sizes *outside* the cloud macro- and microphysics subcycles (Figure 6, right column);

– The cooling, drying, and cloud fraction decreases in the *tropical middle and upper troposphere* (Figure 6, left column) are attributable to shortened $\Delta t_{\mathrm{macmic}}$ (Figure 6, middle column) ;

– The decreases in cloud fraction in the *mid-latitude near-surface layers* are also attributable to shortened $\Delta t_{\mathrm{macmic}}$ (Figure 6j and k).

Geographical distributions of high-cloud and low-cloud fraction changes are shown in Figure 7. The corresponding LW, SW, and total CRE changes are shown in Figure 8. Consistent with the signatures seen in the pressure-latitude cross-sections in Figure 6, one can see the major impact of $\Delta t_{\mathrm{macmic}}$ on high-cloud fraction (Figure 7 top row) and LWCRE (Figure 8 top row). The step sizes outside the cloud macro- and microphysics subcycles play a major role in affecting the low-cloud fraction (Figure 7 second row) and SWCRE (Figure 8 second row). Although reductions in the various step sizes all lead to weakening of







**Figure 6.** Differences in 10-year mean, zonally averaged air temperature (T), specific humidity (Q), relative humidity (RH), and cloud fraction (f) between various simulations. Left column: v1_All_Shorter - v1_CTRL; middle column: v1_MacMic_Shorter - v1_CTRL; right column: v1_All_Except_MacMic_Shorter - v1_CTRL. Statistically insignificant differences are masked out in white. The simulation setups are described in Section 2.3, Table 1, and Table A1. The flowcharts can be found in Figures 1, A1, and A2a.



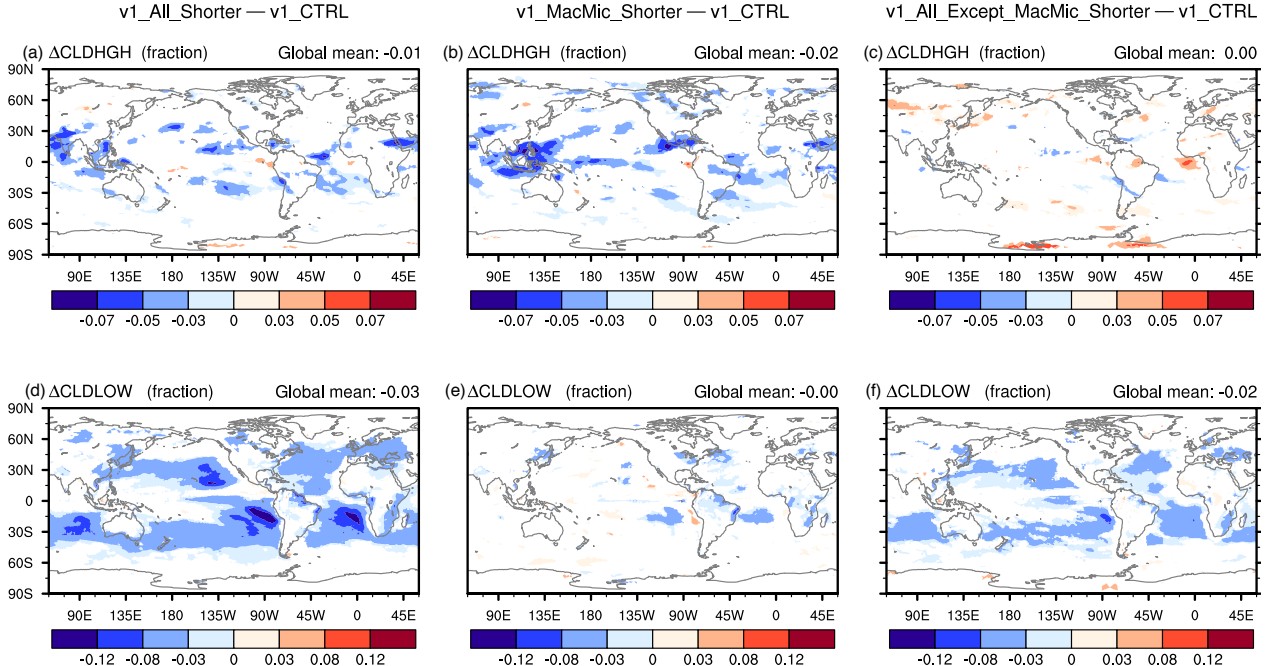

**Figure 7.** Geographical distribution of 10-year mean differences in high-cloud fraction (CLDHGH, upper row) and low-cloud fraction (CLD-LOW, bottom row). Left column: differences between v1_All_Shorter and v1_CTRL revealing the impact of shortening all major time steps listed in Eqs. (1)-(4); middle column: differences between v1_MacMic_Shorter and v1_CTRL revealing the impact of shortening time steps for the subcycled cloud macro- and microphysics parameterizations; right column: differences between v1_All_Except_MacMic_Shorter and v1_CTRL revealing the impact of shortening step sizes outside the cloud macro- and microphysics subcycle. Statistically insignificant results are masked out in white. The simulation setups are described in Section 2.3, Table 1, and Table A1. The flowcharts can be found in Figures 1, A1, and A2a.

both LWCRE and SWCRE, the total CRE changes seen in Figure 8g are dominated by the SW changes attributable to reduced low-cloud fractions associated with shorter time steps outside the cloud macro- and microphysics subcycles (Figures 8f and Figure 7f).

These results are rather counter-intuitive at first glance. Since the topical upper troposphere is strongly affected by deep convection and the resulted detrainment of water vapor and cloud condensate, one might have assumed the sensitivities in these regions to be caused primarily by step sizes associated with the deep convection parameterization – or dynamics and other processes that introduce atmospheric instability which in turn triggers deep convection. Yet the results shown in Figures 6 and 7 suggest that the cloud fraction decreases in these regions are caused by shortening $\Delta t_{\mathrm{macmic}}$, the step size used by turbulence, shallow convection, and stratiform clouds. A separate study has found evidence that the sensitivities in the tropical upper troposphere have to do with the representation of ice cloud microphysics in EAM. The details will be reported in a different



**Figure 8.** As in Figure 7, but showing the longwave (LW, top row), shortwave (SW, middle row), and total (bottom row) CRE.





paper and hence not elaborated here. The link between mid-latitude near-surface clouds and $\Delta t_{\mathrm{macmic}}$ is unclear and needs further exploration.

The results shown in Figures 6–8 lead to the conclusion that step sizes used outside the cloud macro- and microphysics subcycles are more impactful than the step size of the subcycles (i.e., $\Delta t_{\mathrm{macmic}}$) in terms of the sensitivities they cause in cloud fraction and total CRE of the subtropical low clouds. This is also counter-intuitive, and suggests that the low-cloud

differences between v1_All_Shorter and v1_CTRL reflect responses of the subcycled processes to changes in the atmospheric environment passed into the subcyles. In other words, hypothesis 2 is valid for the subtropical low clouds. Next, we demonstrate in Section 4.2 that those low-cloud changes are associated with changes in the thermodynamic (instead of dynamic) features of the atmospheric environment. In Section 4.3, additional sensitivity experiments are presented to further attribute these changes to specific processes and step sizes in the rest of EAMv1.

**4.2 Dynamic versus thermodynamic responses of the subtropical climate**

Large-scale subsidence is one of the key features of the subtropical climate. To find out whether the reduced low-cloud fraction and weaker CRE in v1_All_Shorter are associated with weakened subsidence, the method from Bony et al. (2004) is used to compare the dynamic and thermodynamic components of the low-cloud changes.

We first examined the geographical distribution of grid-resolved vertical velocity $\omega$ at 500 hPa; the differences among the

various simulations discussed so far appeared to be rather small and statistically insignificant and hence are not shown here. This conclusion can also be inferred from the frequency of occurrence of 500 hPa $\omega$, denoted here as $P_\omega$, shown in Figure 9. $P_\omega$ here is diagnosed using monthly-mean grid-point-by-grid-point $\omega$ values in the latitude band of 35°S to 35°N. The solid black line in Figure 9 is $P_\omega$ in v1_CTRL; the dashed color lines show differences in $P_\omega$ between other simulations and v1_CTRL. The differences appear to be close to zero compared to $P_\omega$ in v1_CTRL.

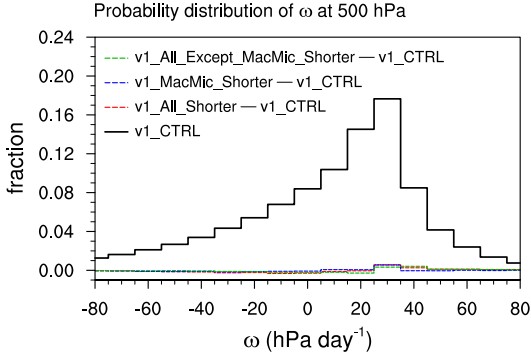

**Figure 9.** Frequency of occurrence of circulation regimes defined by monthly-mean 500 hPa vertical velocity ($\omega$) in the latitude band of 35°S–35°N. Solid black line shows the probability distribution in v1_CTRL. Dashed color lines show differences in the probability distribution between other simulations and v1_CTRL.



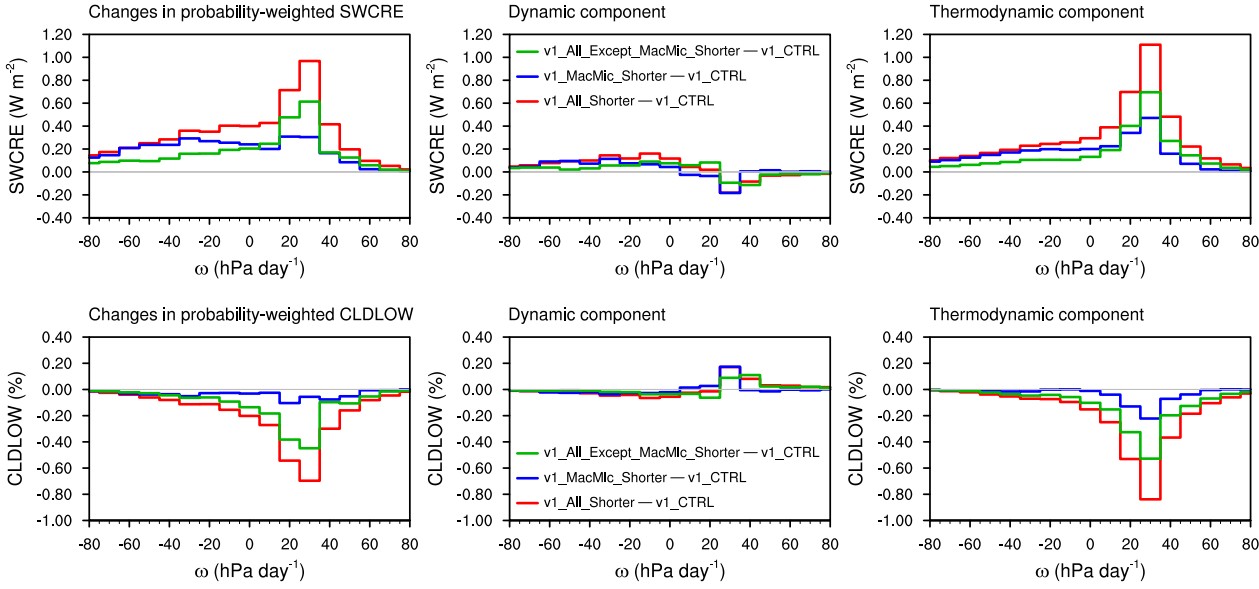

**Figure 10.** Left column: changes in the probability-weighted SWCRE (upper row) and low-cloud fraction (lower row) in circulation regimes characterized by grid-resolved 500 hPa $\omega$ (cf. definitions in Eqs. 5 and 6). Middle and right columns: the dynamic and thermodynamic components of the changes (cf. Eq. 6). Details of the analysis can be found in Section 4.2.

We then followed Bony et al. (2004) and defined circulation regimes using monthly mean $\omega$. For a circulation regime associated with $\omega$ values between $\omega_1$ and $\omega_2$, we refer to the integral of a generic physical quantity $\psi$ weighted by the probability density function $p(\omega)$ as the probability-weighted $\psi$, i.e.,

$$\psi_{(\omega_1,\omega_2)} = \int_{\omega_1}^{\omega_2} \psi\, p(\omega)\, d\omega\,. \tag{5}$$

Following Bony et al. (2004), changes in the probability-weighted $\psi$ can be decomposed as follows:

$$\Delta\psi_{(\omega_1,\omega_2)} \approx \underbrace{\int_{\omega_1}^{\omega_2} \psi\, \Delta[p(\omega)]\, d\omega}_{\text{dynamic}} + \underbrace{\int_{\omega_1}^{\omega_2} (\Delta\psi)\, p(\omega)\, d\omega}_{\text{thermodynamic}} + \underbrace{\int_{\omega_1}^{\omega_2} (\Delta\psi)\, \Delta[p(\omega)]\, d\omega}_{\text{covariation}} \tag{6}$$

In Figure 10, we present changes in the probability-weighted low-cloud fraction and SWCRE in the left column and the decomposition in the middle and right columns. The results suggest that the low-cloud and SWCRE changes in regions associated with subsidence can be attributed primarily to the thermodynamic responses of the model atmosphere instead of vertical velocity changes.





### 4.3 Further attribution of subtropical low-cloud changes


In earlier sections, it has been shown that the reduction in subtropical marine low-cloud fraction and CRE in v1_All_Shorter are caused primarily by the use of shorter time steps for model components outside the cloud macro- and microphysics subcycles and for the coupling (i.e., information exchange) between the subcycles and the other components. We now make an attempt to refine the granularity of the attribution. Additional sensitivity experiments are discussed in this subsection and summarized

as group III in Tables 1 and A1. Schematics of the new simulations are shown in Figure A2b and Figure 12. An overview of the attribution process is provided in Figure 2 with pointers to the figures.

#### 4.3.1 Resolved dynamics and radiation

A simulation was configured similar to v1_All_Except_MacMic_Shorter but with some aspects of the model step sizes returned to values closer to v1_CTRL. Specifically, the horizontal advection time step ($\Delta t_{\mathrm{adv}}$) in the dynamical core was set

to 5 min and the frequency of the radiation calculation $\Delta t_{\mathrm{rad}}$ was set to 60 min, both the same as in v1_CTRL. On the other hand, $\Delta t_{\mathrm{CPLmain}}$ (which controls the step sizes of deep convection, gravity wave drag, various aerosol processes, exchanges with surface models, and the coupling between these processes as well as between physics and dynamics) was set to the smaller value of 5 min. The subcycled cloud macro- and microphysics used time steps of 5 min as in v1_CTRL but was coupled with the rest of the model more frequently (i.e., every 5 min). Because of the required relationship among $\Delta t_{\mathrm{remap}}$,

$\Delta t_{\mathrm{adv}}$ and $\Delta t_{\mathrm{CPLmain}}$ (cf. Section 2), this new simulation ended up using a value of $\Delta t_{\mathrm{remap}}$ in between what were used in v1_CTRL and v1_All_Except_MacMic_Shorter, but the impact is expected to be small. This new experiment is referred to as **v1_CPL+DeepCu_Shorter** (cf. Table 1 and Figure A2b); it differs from v1_All_Except_MacMic_Shorter in its larger (i.e., reverted to EAMv1 default) values of $\Delta t_{\mathrm{adv}}$ and $\Delta t_{\mathrm{rad}}$.

The CRE differences between this new experiment and v1_CTRL, displayed in the left column of Figure 11, show strong

similarity to the rightmost panel of Figure 8. In contrast, the impact of dynamics and radiation time steps, indicated by the CRE differences between v1_All_Except_MacMic_Shorter and this new experiment (Figure 11, right column), appears to have small and mostly insignificant magnitudes. The small yet systematic differences in LWCRE in the Southern Hemisphere midlatitudes (Figure 11b) indicate a shift in the location of the storm tracks, but no systematic signals are seen in the lower latitudes. Therefore we conclude that the impact of dynamics and radiation time steps on subtropical clouds is small, at least

in the context of the currently used process ordering and splitting/coupling methods.

#### 4.3.2 Coupling between cloud macro/microphysics and other processes

We are now left with two time step sizes to explore, $\Delta t_{\mathrm{CPLmain}}$ and $\Delta t_{\mathrm{deepCu}}$. As explained in Section 2 and illustrated by color coding in the flowcharts, these two step sizes have the same value in EAMv1, and the single $\Delta t_{\mathrm{CPLmain}}$ also controls the coupling frequency among the majority of the parameterizations as well as between physics and dynamics. This makes further

attribution somewhat difficult unless changes are made to the model source code. Nevertheless, our exploration revealed that the coupling between the subcycled cloud parameterizations and the rest of the model was impactful. This can be demonstrated





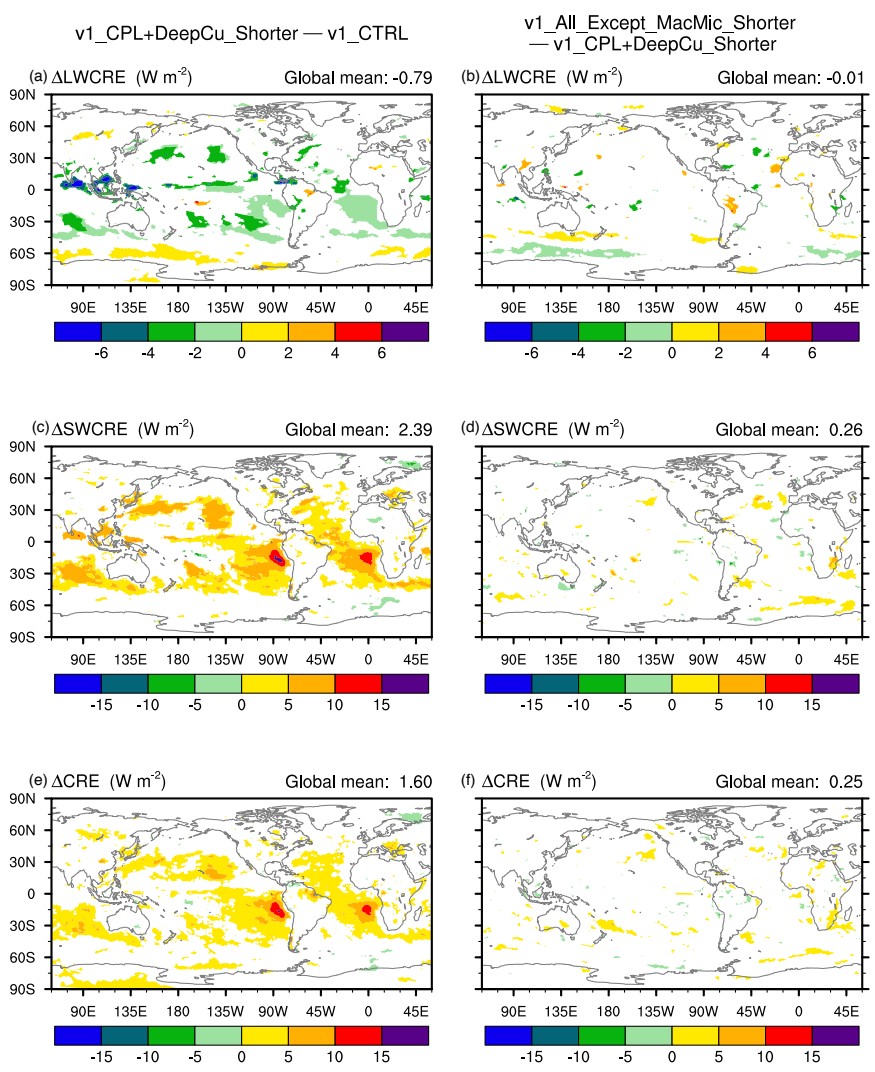

**Figure 11.** Attribution of the 10-year mean CRE differences shown in the rightmost column of Figure 8. Left: CRE differences between v1_CPL+DeepCu_Shorter and v1_CTRL revealing the impact of step sizes used for process coupling/splitting and deep convection. Right: CRE differences between v1_All_Except_MacMic_Shorter and v1_CPL+DeepCu_Shorter revealing the impact of dynamics and radiation time steps. White indicates statistically insignificant differences. The simulation setups are summarized in Tables 1 and A1. The flowcharts can be found in Figures 1 and A2.



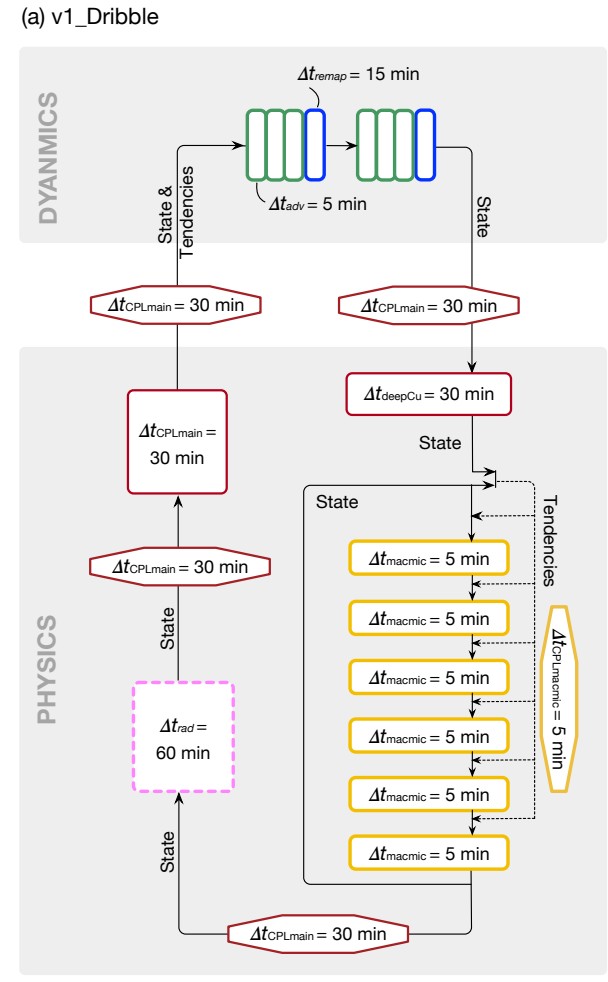

**Figure 12.** As in Figure 1b but for simulation v1_Dribble (cf. Tables 1 and A1).

using the simulation **v1_Dribble** which is similar to v1_CTRL but employs a revised process coupling strategy depicted in Figure 12. In the revised scheme, the atmospheric temperature, specific humidity, as well as cloud liquid and ice concentrations that are passed to the first cloud macro/microphysics subcycle are no longer the values updated by the preceding processes.

Instead, the older snapshot saved after the last (6th) 5 min cloud macro/microphysics subcycle in the previous main coupling time step of $\Delta t_{\mathrm{CPLmain}}$ is provided together with the corresponding total tendencies caused by all processes outside the subcycles. Before each subcycle, those tendencies are used to update the atmospheric state using a step size of $\Delta t_{\mathrm{macmic}} = 5$ min. This "dribbling" of tendencies is conceptually similar to the physics-dynamical coupling used by EAMv1's dynamical core for temperature, winds and surface pressure (Zhang et al., 2018; Rasch et al., 2019); it can be viewed as an example of the

sequential-tendency splitting method defined in Donahue and Caldwell (2018). To help distinguish this "dribbling" from the original sequential splitting, we introduced the notation $\Delta t_{\mathrm{CPLmacmic}}$ in Eq. (2) as well as in Table 1 and Figure 2. "Dribbling"



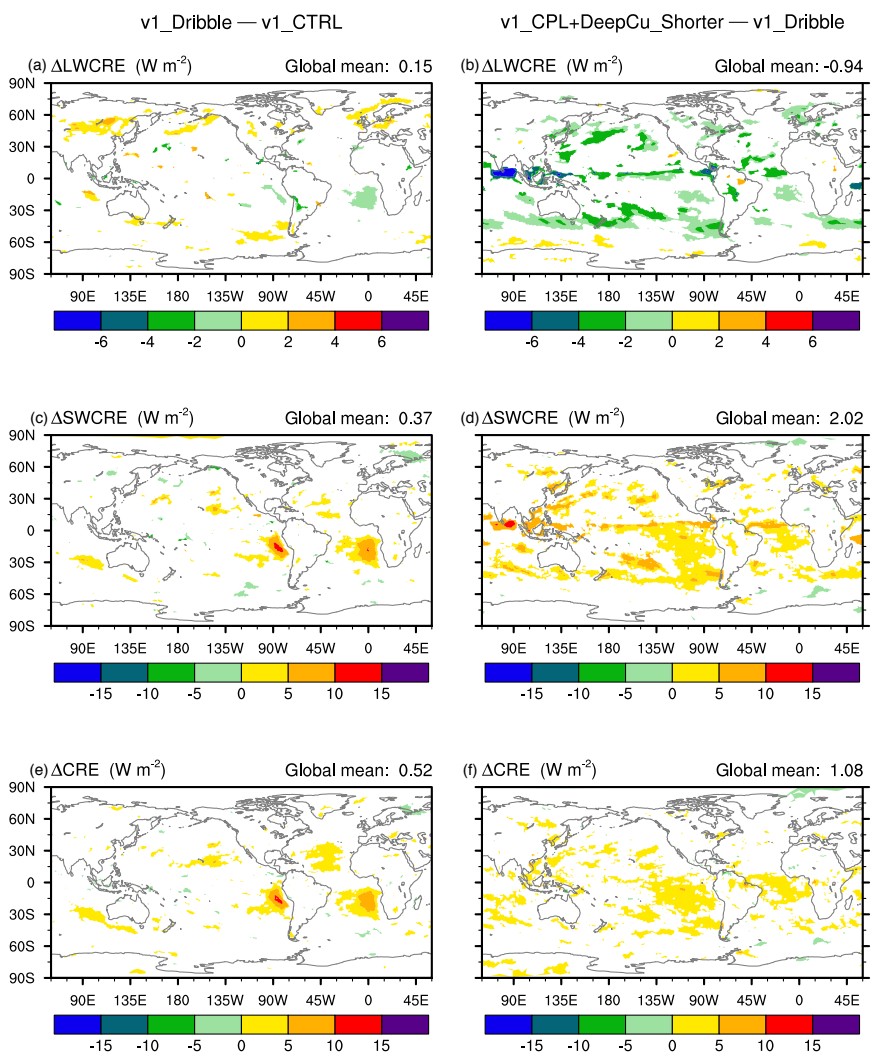

**Figure 13.** Attribution of the 10-year mean CRE differences shown in the left column of Figure 11. Left: differences between v1_Dribble and v1_CTRL revealing the impact of coupling between the subcycled cloud macro-/microphysics and the rest of EAM. Right: differences between v1_CPL+DeepCu_Shorter and v1_Dribble revealing the impact of step sizes used by various other parameterizations (deep convection, gravity wave drag, various aerosol processes) and the coupling among them. White indicates statistically insignificant differences. The simulation setups are summarized in Tables 1 and A1. Flowcharts are shown in Figures 1, 12, and A2b.





provides a more frequent coupling from the perspective of the subcycled cloud macro/microphysics, while the feedback to the processes outside the subcycles still occurs at longer intervals of $\Delta t_{\text{CPLmain}}$. A detailed explanation of the motivation for this "dribbling" and an in-depth analysis of its impact on the atmospheric water budget will be the topic of a separate paper. Here

we only show the CRE differences between the two simulations v1_Dribble and v1_CTRL (Figure 13, left column), in which weakened SWCRE and total CRE are found over the eastern parts of the subtropical oceans, especially in the regions with frequent occurrences of stratocumulus decks in the default EAMv1 model. This suggested that the strongest local reduction of SWCRE and/or total CRE seen earlier in simulations v1_All_Shorter (Figure 8d and g), v1_All_Except_MacMic_Shorter (Figure 8f and i), and v1_CPL+DeepCu_Shorter (Figure 11c and e) are primarily attributable to more frequent coupling between

the subcycled cloud macro/microphysics and the rest of EAM.

### 4.3.3   Deep convection

The simulation v1_Dribble introduced in Section 4.3.2 employed tighter coupling between the subcycled cloud macro/microphysics parameterizations but otherwise used the same step sizes as in the default model. The simulation v1_CPL+DeepCu_Shorter introduced earlier in Section 4.3.1 not only had this tighter coupling but also used 6 times shorter step sizes for other miscel-

laneous parameterizations and their interactions, including deep convection, land surface, gravity wave drag, aerosol sedimentation and dry deposition, as well as aerosol microphysics (but not activation). Therefore the differences in results between v1_CPL+DeepCu_Shorter and v1_Dribble provides an estimate of the impact of these other miscellaneous time steps and coupling frequencies.

Figure 13 shows the 10-year mean differences in CRE, revealing weaker LWCRE and SWCRE along the equatorial ITCZ

(Inter Tropical Convergence Zone, where deep convection is important) and in the subtropics and equatorward flanks of the storm tracks. The LW and SW changes largely cancel each other along the equator and near the storm tracks, leaving differences in the net CRE visible only in the trade cumulus regions. The signatures of a net cancellation in LWCRE and SWCRE along the ITCZ provide hints that there is a change in behavior in the deep convection regime. We speculate there are three possible reasons for a deep convection parameterization to be sensitive to model step size: (i) temporal truncation errors associated

with $\Delta t_{\text{deepCu}}$, (ii) the ratio between $\Delta t_{\text{deepCu}}$ and $\tau$ where $\tau$ is the prescribed (fixed) time scale for releasing the convective available potential energy (CAPE) in the Zhang and McFarlane (1995) scheme, and (iii) the coupling between deep convection and other processes which is determined essentially by the ratio of $\Delta t_{\text{deepCu}}$ to $\Delta t_{\text{CPLmain}}$; this ratio is 1 in the current EAM and in all simulations in this paper.

Williamson (2013) pointed out that convective parameterizations designed to remove instability and supersaturation on

assumed constant $\tau$ are constrained by $\Delta t_{\text{deepCu}}$ in how much work such parameterizations can do in each time step. In contrast, large-scale condensation parameterizations designed to completely remove supersaturation within every time step are unconstrained, with implications for the column instability and depth of convection that in turn affects the resolved dynamical response. This difference in characteristic behavior can affect the simulated interactions between dynamics, deep convection and the stratiform cloud processes. Williamson (2013) showed that this time-step-time-scale issue ($\Delta t_{\text{deepCu}}/\tau$ issue) could

explain the occurrence of intense truncation-scale storms in high-resolution simulations conducted with the Community Atmo-





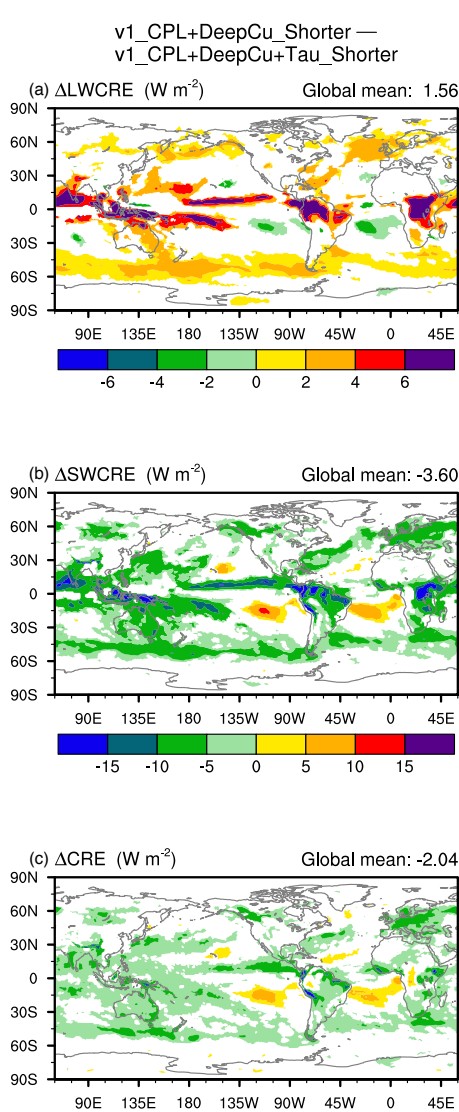

**Figure 14.** 10-year mean CRE differences between v1_CPL+DeepCu_Shorter and v1_CPL+DeepCu+Tau_Shorter reveal the impact of a reduced ratio of $\Delta t_{\mathrm{deepCu}}/\tau$ without model step size changes. White indicates statistically insignificant differences. The simulation setups are summarized in Tables 1 and A1. The two simulations correspond to the same flowchart shown in Figure A2b.





sphere Model Version 4 (CAM4). Other studies, e.g., Mishra et al. (2008), Mishra and Srinivasan (2010), Mishra and Sahany (2011), and Yu and Pritchard (2015), have also shown model sensitivities to $\Delta t_{\mathrm{deepCu}}$ and/or $\tau$.

The impact of a smaller $\Delta t_{\mathrm{deepCu}}/\tau$ ratio alone, i.e., without changes in parameterization step size or process coupling frequency, can be derived by comparing the simulation v1_CPL+DeepCu_Shorter already introduced earlier and a new sim-

ulation **v1_CPL+DeepCu+Tau_Shorter** (cf. Table 1 and Figure 2, Figure A2b). The latter was configured to be identical to the former except for a different value of $\tau$. In other words, the $\Delta t_{\mathrm{deepCu}}/\tau$ ratio is 1/6 of its default value in simulation v1_CPL+DeepCu_Shorter because of the shorter $\Delta t_{\mathrm{deepCu}}$, while the ratio was returned to its default value in EAMv1 in the new simulation v1_CPL+DeepCu+Tau_Shorter because $\tau$ was also shortened proportionally. The CRE differences between v1_CPL+DeepCu_Shorter and v1_CPL+DeepCu+Tau_Shorter, shown in Figure 14, indicate the impact of a smaller

$\Delta t_{\mathrm{deepCu}}/\tau$ when model time steps and coupling frequencies remain unchanged. A smaller ratio decreases the relative importance of the convective parameterization per time step and therefore amplifies the role of the stratiform cloud parameterizations (and associated Hadley circulation); this makes the (positive) LWCRE more positive and the (negative) SWCRE more negative in convective regions. In other words, the strengthened amplitude of both LW and SW CRE in the ITCZ seen in Figure 14 is consistent with the response described in Williamson (2013).

Comparing Figure 14 with the right column of Figure 13, however, we see different signs and patterns of changes in the ITCZ. Recall that in the right column of Figure 13, the simulation v1_CPL+DeepCu_Shorter used a $\Delta t_{\mathrm{deepCu}}$ 6 times shorter than in v1_Dribble; this not only led to a smaller $\Delta t_{\mathrm{deepCu}}/\tau$ ratio but also implied that the deep convection parameterization was invoked 6 times more frequently and interacted with the resolved dynamics and other parameterized processes 6 times more frequently. The discrepancies between the right column of Figure 13 and Figure 14 suggest that the more frequent

invocation of deep convection and more frequent coupling with other processes lead to consequences that compensate (in fact overcompensate) the impact of a smaller $\Delta t_{\mathrm{deepCu}}/\tau$. In other words, the overall responses of the annual mean CREs to a shortened $\Delta t_{\mathrm{deepCu}}$ is inconsistent with the time-step-time-scale discussion in Williamson (2013). Clearly, more work is needed in understanding the process interactions in the model and how these interactions respond to changes in step sizes and coupling frequencies.

**5 Conclusions**

This study evaluated the strength of time step sensitivities in 10-year present-day climate simulations conducted with the EAMv1 atmospheric model at 1-degree horizontal resolution. A proportional, factor-of-6 reduction of time step size in major components of the model was found to result in changes in the long-term mean climate that were significant both statistically and physically. A systematic warming was found in the low-latitude areas in the near-surface levels and a systematic cooling

was seen aloft. Drying throughout the troposphere was found to be accompanied by sizable cloud fraction decreases; the primary signatures of the latter occurred at most latitudes in the upper troposphere, in the subtropical lower troposphere, and in the mid-latitude near-surface layers. In terms of geographical distribution, the most pronounced annual mean changes are the decreases in total cloud cover and CRE over the subtropical marine stratocumulus and trade cumulus regions.





The comparison of model results with a comprehensive set of observational data indicated that the changes caused by step size reduction led to a degradation in model fidelity, in terms of both the global mean statistics and the geographical distributions. Although this is not surprising given the careful tuning EAMv1 has gone through, the compensation of time integration error by parameter tuning or by other sources of model error is undesirable, which also implies the need for additional tuning to reach a new compensation when model time steps are shortened for high-resolution simulations. It would be more desirable to identify time step algorithms with numerical errors that are small enough that the simulation fidelity is insensitive to reasonable variations in step size; that is, so that the simulation quality is determined by physical understanding (or lack of it).

In order to provide clues for future efforts on reducing time-stepping errors in EAM, additional simulations were conducted to tease out some of the sources of time step sensitivities seen in EAMv1. Most of those simulations made use of flexible choices of time step sizes currently available in various subsets of EAM's components. One of the simulations used an alternate numerical scheme to couple the collectively subcycled shallow cumulus and stratiform cloud macro/microphysics parameterizations with the rest of EAM. Another simulation used a different value for the CAPE removal time scale in the deep convection parameterization to investigate the impact of time-step-time-scale ratio.

Analysis of the results focused mostly on annual mean cloud fraction and CRE. We found that the most notable sensitivity in the simulations was changes in total cloud cover and CRE in the subtropical marine stratocumulus and trade cumulus regimes. Surprisingly, we found that this sensitivity was not attributable to the step size used for treating some of the most important processes in those regimes (turbulence, and cloud macro- and microphysical processes), but rather to the strategy used to couple those processes to other components of the model. On the other hand, the step size of the cloud macro- and microphysics subcycles had quite an important impact on cloud fraction at most latitudes in the upper troposphere as well as in the mid-latitude near-surface layers. Additional simulations and analysis revealed that the deep convection parameterization and its coupling with other processes significantly affects trade cumulus. Impacts of the step sizes used by the dynamical core and radiation are small. In Figure A4, we have reorganized some of the CRE difference plots presented in earlier sections: a different panel layout is used to facilitate a direct comparison of the impacts of step sizes used by different model components. Recent follow-up or independent studies have provided insights into the impact of process coupling on marine stratocumulus clouds and the impact of macro/microphysics time step on ice cloud formation. Those results will be reported in separate papers. The mechanisms under the other sensitivities shown in the figure still need to be investigated.

Using the analysis method of Bony et al. (2004), we found that the subtropical low-cloud changes were primarily local, thermodynamic responses of the model atmosphere while the impact of circulation (vertical velocity) changes was very small. This conclusion has practical implications for follow-up investigations: since circulation changes are negligible and local cloud processes are fast, it should be feasible to use nudged 1-year simulations (Kooperman et al., 2012; Zhang et al., 2014; Sun et al., 2020) or even ensembles of few-day simulations (Xie et al., 2012; Ma et al., 2013; Wan et al., 2014) to help carry out further investigations at process level and meanwhile keep the numerical experiments computationally economical.

While this study focused on one specific AGCM, we would like to advocate similar exercises be carried out with other models as well. Based on considerations of computational cost, numerical models used for operational weather forecast and climate





research generally tend to use the longest step sizes that would provide satisfactory results. To which extent the key features
of those results would depend on the chosen step sizes, however, is not always clear. If a time step sensitivity quantification
exercise like ours presented in Section 3 reveals strong sensitivities, that would provide a motivation to understand the causes of
the sensitivities, and, in the next step, revise the numerical methods to provide higher accuracy without substantially increasing
the computational cost. Root causes of time step sensitivities in comprehensive atmospheric models like EAM can be very
challenging to identify due to the complex interactions and feedbacks between physical processes. Our study shows that by
making use of the time-stepping subcycles in a model and implementing alternate methods for process coupling, it is possible
to narrow down the culprit of certain sensitivities and provide clues for further investigation and improvements.

*Code availability.* The EAMv0 and v1 source codes and run scripts used in this study can be found on Zenodo at https://doi.org/10.5281/
zenodo.4118705.

## Appendix A: Additional table and figures

Table A1 documents the namelist settings used in the EAMv0 and EAMv1 simulations presented in this paper to help interested
readers reproduce our experiments. Figures A1 and A2 show flowcharts for sensitivity experiments conducted with EAMv1
that are not shown in Figures 1 and 12. Figure A3 presents the geographical distribution of SWCRE biases in v0_CTRL and
v1_CTRL to show that the two models have different characteristics in the spatial distribution of model biases (cf. Section 3.2).
In Figure A4, the CRE differences between various simulations discussed in Sections 3 and 4 are presented in a reorganized
layout to help highlight the conclusions of our sensitivity attribution effort.

**Table A1.** Namelist setup used by the simulations listed in Table 1.

| Group | Simulation | Description | Flowchart | Namelist variables and their values | | | | | |
| --- | --- | --- | --- | --- | --- | --- | --- | --- | --- |
| | | | | se_nsplit | rsplit | dtime | cld_macmic_num_steps | iradsw, iradlw | zmconv_tau |
| 0 | v0_CTRL | Sect. 2.2 | - | 2 | 3 | 1800 | N/A | -1, -1 | 3600 |
| I | v1_CTRL | Sect. 2.1 | Fig. 1 | 2 | 3 | 1800 | 6 | -1, -1 | 3600 |
| I | v1_All_Shorter | Sect. 2.3 | Fig. A1a | 2 | 3 | 300 | 6 | 2, 2 | 3600 |
| II | v1_MacMic_Shorter | Sect. 4.1 | Fig. A1b | 2 | 3 | 1800 | 36 | -1, -1 | 3600 |
| II | v1_All_Except_MacMic_Shorter | Sect. 4.1 | Fig. A2a | 2 | 3 | 300 | 1 | 2, 2 | 3600 |
| III | v1_CPL+DeepCu_Shorter | Sect. 4.3.1 | Fig. A2b | 1 | 1 | 300 | 1 | -1, -1 | 3600 |
| III | v1_CPL+DeepCu+Tau_Shorter | Sect. 4.3.3 | Fig. A2b | 1 | 1 | 300 | 1 | -1, -1 | 600 |
| III | v1_Dribble | Sect. 4.3.2 | Fig. 12 | 2 | 3 | 1800 | 6 | -1, -1 | 3600 |





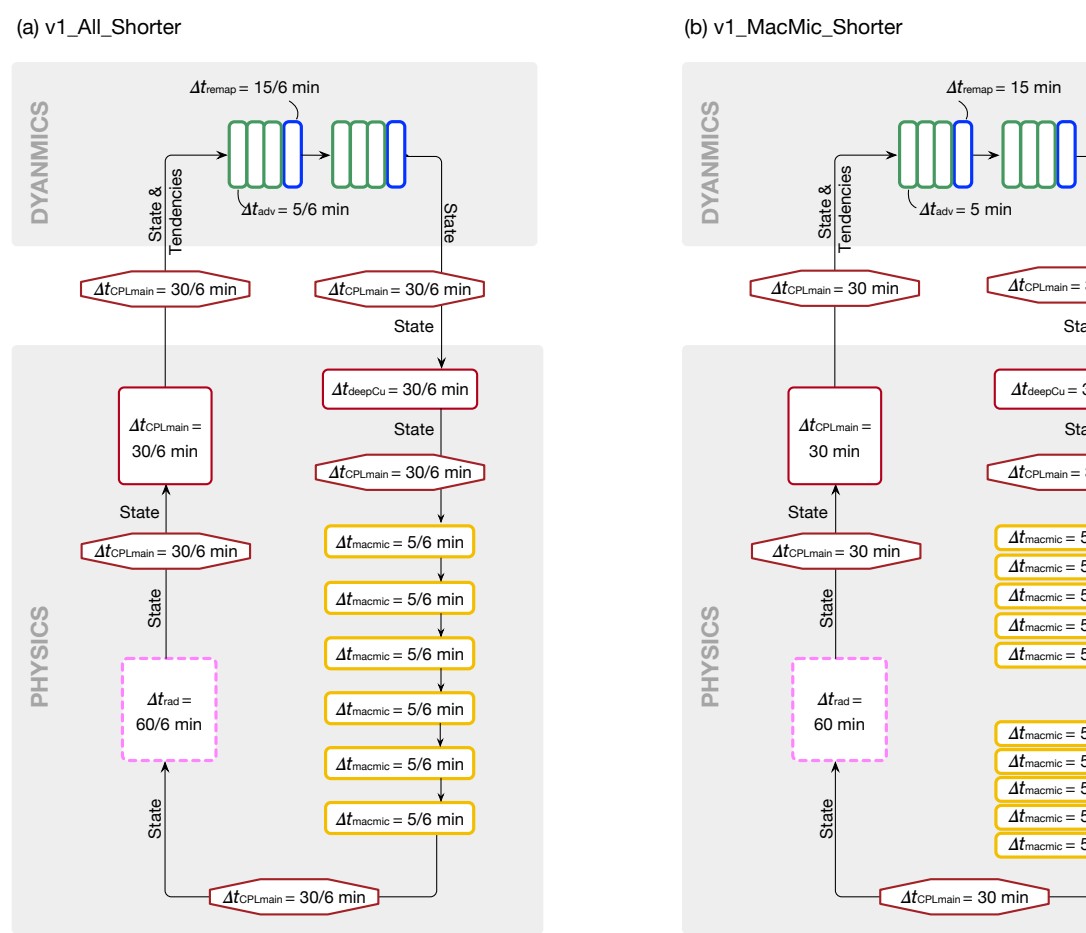

**Figure A1.** As in Figures 1b and 12 but for two additional simulations listed in Tables 1 and A1.





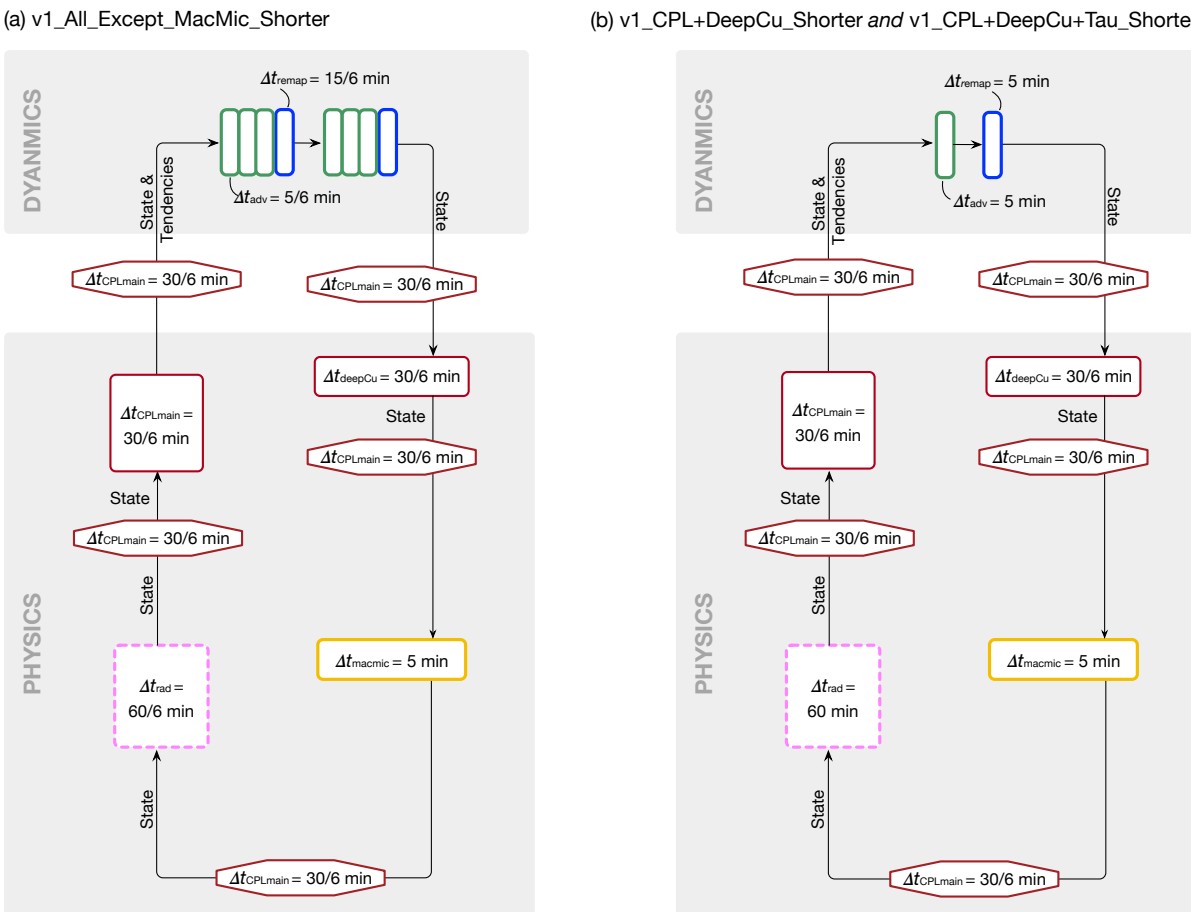

**Figure A2.** As in Figures 1b and 12 but for three additional simulations listed in Tables 1 and A1.

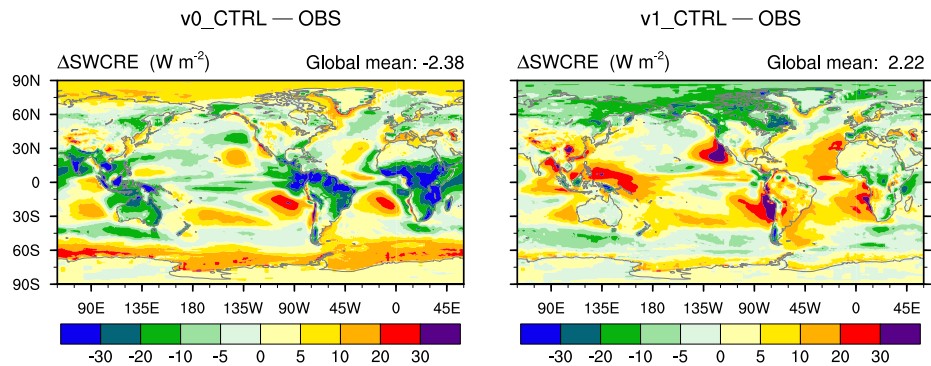

**Figure A3.** 10-year mean differences in SWCRE between simulations conducted with v0_CTRL (left) or v1_CTRL (right) and the 2000-2010 averages from CERES-EBAF.





**Figure A4.** Attribution of the 10-year mean CRE differences between v1_All_Shorter and v1_CTRL (first row) to various components of EAMv1 (lower rows). Left column: LWCRE; middle column: SWCRE; right column: total CRE. White indicates statistically insignificant differences. The attribution process is summarized in Figure 2. The simulation setups are summarized in Tables 1 and A1.





*Author contributions.* HW initiated this study and designed the sensitivity experiments with input from the coauthors. HW conducted the EAMv0 simulation. SZ carried out the EAMv1 simulations and processed all the model output. HW and SZ led the analysis of the results and the other authors provided feedback. HW wrote the first draft of the manuscript and led the subsequent revisions. All coauthors contributed to the revisions.

*Competing interests.* The authors declare no competing interests.

*Acknowledgements.* The authors thank Kai Zhang for helpful discussions during this study and for his comments on earlier versions of the paper. This work was supported by the U.S. Department of Energy (DOE), Office of Science, Office of Biological and Environmental Research (BER) via the Scientific Discovery through Advanced Computing (SciDAC) program. Computing resources were provided by the National Energy Research Scientific Computing Center (NERSC), a DOE Office of Science User Facility operated under Contract No. 475 DE-AC02-05CH11231. Pacific Northwest National Laboratory is operated for DOE by Battelle Memorial Institute under contract DE-AC06-76RLO 1830.





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
