# Peer review of "Quantifying and attributing time step sensitivities in present-day climate simulations conducted with EAMv1"

_Geoscientific Model Development, 2020_

## Referee Comment (RC1) · Andrew Barrett (Referee) · 27 Nov 2020

This paper addresses the important and until-now under studied impact of changing time-stepping and physics-dynamics coupling methodologies within numerical models of the atmosphere. In this work, the Energy Exascale Earth System Model is used to investigate the impact of a factor of 6 shortening in the time step for 10-year duration simulation of the present (year 2000) climate. The shortening of the time step reduces errors associated with truncation errors in time and also allows a tighter coupling of the physical processes which are parameterized separately within the model (such as dynamics, clouds, convection, and radiation). These changes have a small to moderate impact on the overall radiative balance of the model resulting in a warming of the lower troposphere (+0.5K), cooling of the upper troposphere/lower stratosphere (-0.5K), a decrease in low level relative humidity (1%, locally up to 10%) and a reduction of low level clouds, particlcuarly in subtropical stratocumulus & shallow-cumulus regions. These changes were determined to result from local thermodynamics changes rather than changed to the circulation.

The authors then further investigate which part of the coupling is responsible for these changes through a series of experiments where only a subset of the model's parameterizations use a shortened timestep. The conclusion of these experiments is that differences come from the calculation of the cloud marcophysics+microphysics, but are not caused by sensitivities within the cloud microphysics+microphysics calculations; rather they are caused by changes to the input values to the cloud macrophysics+microphysics parameterization, which result from changing time step length in other parts of the model physics (e.g. larger temperature increments due to longer radiation or dynamics time steps). In an attempt to study this effect, they briefly introduce the "dribble" method of substepping the cloud microphysics with a shorter time step and providing partial updates of the model prognostic variables to each substep based on their change since the last time step.

-

I find the study to be well constructed, the paper well written and the figures generally clear and justifying inclusion in the paper. I find the results convincing and have no significant concerns regarding the method nor the scientific conclusions of the paper. However, I have a few suggestions for making the paper clearer and improving the clarity of the messages within it. In my opinion the paper is already worthy of publication and my suggestions are intended to improve the clarity of some parts of the paper.

-

1) I suggest including quantification of the changes to temperature, RH and clouds in

both the abstract and conclusions so that the scale of the time step sensitivity is easily findable.

2) I do not see the value of including information about EAMv0 model output in the paper. The authors should think about whether it is really important to include the description of this model version and the changes in biases between v0 and v1. I didn't find that this information added (or detracted) from my understanding of the time step sensitivity.

3) I found the schematic figure 2 incredibly valuable - showing which sets of simulations I should compare to see what effect. However, the text-based descriptions of the different time-stepping (coupling) strategies I found confusing and made my own notes/sketches when reading the paper. Only at the end of reading the paper did I find the schematics in figures A1 & A2. In this case the figures are much easier to compare than long sections of text and I would therefore suggest adding more explicit references to the explanatory figures at the beginning of each block of text, rather than at the end. (a specific example: more the reference to Figure A2b from line 322 to line 314 (perhaps moving the whole sentence)).

4) The sections 4.3.2 (dribble method) and 4.3.3 (shorter tau in convective param.) are confusing to me. The method is introduced, the results are very briefly summarized and a summary sentence is reached. However, I do not follow which figures should be compared to reach the same conclusions as the authors. Furthermore, the authors do not include any information from these two subsections in their conclusions and therefore I think the meaning of these sections is lost. Either these sections (and the conclusions) should be expanded to include the logic and comparison of the relevant figures, or could be omitted from the paper completely without losing impact (as they are currently difficult to understand - at least by me). You already mention that an additional paper is being prepared about the dribble method, so perhaps that would be the place for a more expansive discussion.

5) The authors state that the cause of the time step sensitivity coming from outside, rather than within, the subcycled physics is "counter-intuitive" (abstract and line 279). However, this is exactly the mechanism for time step sensitivity that I reported in Barrett et al. (2019) from cloud-resolving model simulations (with time step ranges from 1-15 seconds). I agree that it was initially counter intuitive, but is related to the requirement for the cloud parameterizations to react to a large push away from equilibrium by condensation (resulting from cooling imposed by the dynamics and radiation tendencies). I therefore suggest:

5.1) Adding a sentence that this is potentially caused by condensation/cloud formation in the model and the sensitivity does not get removed even with time steps of only a few seconds (which are not achievable in climate simulations), strengthening the argument for the importance of finding a good numerical solution to this problem (perhaps dribbling...)

5.2) Including information in the model description about which part of the model diagnoses the cloud fraction and condensation (i.e. whether this occurs inside or outside of the microphysics+macrophysics and how it is treated within the subcycling setups)

Typos found:

Line 90: this *is* tied to

Line 440: The mechanisms *under* -> *behind*

---

## Referee Comment (RC2) · Anonymous Referee #2 · 21 Dec 2020

Overview

In this manuscript, the authors investigate the sensitivity to model timestep of the mean climate within the EAMv1 model. They then methodically scrutinize the response to variations of the model sub-component timesteps in a series of experiments that are designed to attribute the root cause of the model sensitivity to the individual components.

Perhaps unsurprisingly to those familiar with model development, the largest deviations can be attributed to the parametrizations of clouds and moist convection. Perhaps less predictable is how and where these deviations are - in part due to the direct change

to the sub-timestepping of the microphysics scheme and in part arising from timestep sensitivities of their coupling to other components.

The investigations performed are specific to the EAMv1 atmospheric system and in particular to the specific physics parametrizations, dynamical core and transport formulations and split-explicit temporal coupling. That said, the methodology would translate to other modelling systems and formulations, potentially providing a useful framework for identifying physics or dynamics components that are not well resolved or are poorly formulated for a chosen model timestep length. The method can also help in identifying poorly performing components of the system which might otherwise be hidden or explained away with model tuning. For those without an interest in the specific behaviour of EAM, the most valuable take away message is that developing and employing an approach such as this will help to understand deficiencies and biases when developing weather and climate models.

The manuscript is very well laid out, with a clearly constructed story (with an exception noted in comment 5). I would recommend this article for publication subject to consideration of the minor comments below.

General comments

1) On L80, the authors mention the passing of tendencies between different components of the atmospheric model and provide the example of the physics tendencies being passed to the resolved dynamics. It is implied that there are other such instances. Although these aren't the subject of the investigation, it would be useful to know what the other instances are, or if these are too numerous/complex to list, then to clarify if there are cases where different physics components pass their tendencies to subsequent physics components. In particular, do those physics components (clouds, microphysics, convection) which are the focus of the sensitivity studies have such a dependency? If they do, are there any consequences to this that should be borne in mind when interpreting the conclusions about timestep choices between the dependent

schemes. If they don't, it would be useful to have that made explicit.

2) Figure 1: This is a useful schematic to quickly capture the timestepping process. I'm not sure I can see from this where/when the state is updated though. Is it at every point where State is labelled? Could this be made a bit clearer?

3) On first inspection it is quite surprising how little an impact there is in the subtropical low-cloud that can be directly attributed to the reduction of $\Delta t\_macmic$. However, it is noted in the model overview (L74) that the microphysics includes dynamic substepping and so the timestep used within the microphysics itself may already be shorter than 5 minutes - particularly where numerical stability requires it. While this doesn't negate the conclusions, it might be worth re-stating this feature of the microphysics when discussing this lack of sensitivity. Do the authors have any information regarding the minimum timestep that is used or the number of hydrometeor sedimentation substeps that might add to that understanding?

4) Regarding the sensitivity of the tropical upper troposphere to the cloud macro and microphysics, there are a number of ways in which the cloud microphysics can directly and indirectly influence this region. See for example figure 1 of Hardiman, Steven C., et al. "Processes controlling tropical tropopause temperature and stratospheric water vapor in climate models." Journal of Climate 28.16 (2015): 6516-6535.

5) Section 4.3.3 - Deep convection: This section is at a slight tangent to the rest of the paper and as a result it took me several reads through to be able to absorb the conclusions. Unlike the preceding sections that look to attribute sensitivities, this section attempts to investigate the reason behind the sensitivity of the deep convection to timestep and timescale and then contrasts the findings with the arguments of Williamson (2013). This feels like a half-hearted attempt, with the authors themselves acknowledging more work is needed to fully understand this. I am tempted to suggest removal of this section, however, once I had digested it I did find it interesting and so I would suggest that this goes in an appendix. If the authors do decide to retain the

section (in its current location or elsewhere), then please could they clearly signpost at the start of this section that it is a change in direction from the previous stated aims of the paper and is looking beyond attribution?

Typos

L147: 'Working' not 'Wording'

L268: 'Tropical' not 'Topical'

---

## Author Comment (AC1) · 18 Jan 2021

**Referee comment:** This paper addresses the important and until-now under studied impact of changing time-stepping and physics-dynamics coupling methodologies within numerical models of the atmosphere. In this work, the Energy Exascale Earth System Model is used to investigate the impact of a factor of 6 shortening in the time step for 10-year duration simulation of the present (year 2000) climate. The shortening of the time step reduces errors associated with truncation errors in time and also allows a tighter coupling of the physical processes which are parameterized separately within the model (such as dynamics, clouds, convection, and radiation). These changes have

a small to moderate impact on the overall radiative balance of the model resulting in a warming of the lower troposphere (+0.5K), cooling of the upper troposphere/lower stratosphere (- 0.5K), a decrease in low level relative humidity (1%, locally up to 10%) and a reduction of low level clouds, particluarly in subtropical stratocumulus & shallow-cumulus regions. These changes were determined to result from local thermodynamics changes rather than changed to the circulation.

The authors then further investigate which part of the coupling is responsible for these changes through a series of experiments where only a subset of the model's parameterizations use a shortened timestep. The conclusion of these experiments is that differences come from the calculation of the cloud marcophysics+microphysics, but are not caused by sensitivities within the cloud microphysics+microphysics calculations; rather they are caused by changes to the input values to the cloud macrophysics+microphysics parameterization, which result from changing time step length in other parts of the model physics (e.g. larger temperature increments due to longer radiation or dynamics time steps). In an attempt to study this effect, they briefly introduce the "dribble" method of substepping the cloud microphysics with a shorter time step and providing partial updates of the model prognostic variables to each substep based on their change since the last time step.

I find the study to be well constructed, the paper well written and the figures generally clear and justifying inclusion in the paper. I find the results convincing and have no significant concerns regarding the method nor the scientific conclusions of the paper. However, I have a few suggestions for making the paper clearer and improving the clarity of the messages within it. In my opinion the paper is already worthy of publication and my suggestions are intended to improve the clarity of some parts of the paper.

**Author response:** We greatly appreciate Dr. Barrett's very positive assessment and insightful review. Our responses to his suggestions are detailed below.

**Referee comment:** 1) I suggest including quantification of the changes to temperature,

RH and clouds in both the abstract and conclusions so that the scale of the time step sensitivity is easily findable.

**Author response:** The suggested changes are included in the revised manuscript.

**Referee comment:** 2) I do not see the value of including information about EAMv0 model output in the paper. The authors should think about whether it is really important to include the description of this model version and the changes in biases between v0 and v1. I didn't find that this information added (or detracted) from my understanding of the time step sensitivity.

**Author response:** We agree the v0 results do not provide much help in understanding time step sensitivity in v1, as the parameterizations and the vertical resolution in the two model versions are both substantially different. We chose to, and still would like to, keep the v0 results in the manuscript because this little part is intended for a slightly different audience, namely colleagues who have been developing, tuning, and evaluating EAM with a focus on its fidelity in reproducing the observed climate but who might have not thought much about time step sensitivity before reading our manuscript. The v0 results presented here are intended to provide these colleagues with a quantitative assessment of the relative magnitude of the time step sensitivity compared with the changes in model biases when EAM was evolved from v0 to v1.

**Referee comment:** 3) I found the schematic figure 2 incredibly valuable - showing which sets of simulations I should compare to see what effect. However, the text-based descriptions of the different time-stepping (coupling) strategies I found confusing and made my own notes/sketches when reading the paper. Only at the end of reading the paper did I find the schematics in figures A1 & A2. In this case the figures are much easier to compare than long sections of text and I would therefore suggest adding more explicit references to the explanatory figures at the beginning of each block of text, rather than at the end. (a specific example: more the reference to Figure A2b from line 322 to line 314 (perhaps moving the whole sentence)).

[Figure]

**Author response:** We are glad to hear Dr. Barrett's appreciation of the value of the schematics, and we agree that these are more straightforward to read than the text. Following Dr. Barrett's suggestion and example, we have revised the manuscript and moved references to the schematics to the beginning of each block of text description. We also moved schematics that were originally in the Appendix (i.e., Figures A1 and A2) to the main body of the revised manuscript so that they are placed closer to the text descriptions of the corresponding simulations.

**Referee comment:** 4) The sections 4.3.2 (dribble method) and 4.3.3 (shorter tau in convective param.) are confusing to me. The method is introduced, the results are very briefly summarized and a summary sentence is reached. However, I do not follow which figures should be compared to reach the same conclusions as the authors. Furthermore, the authors do not include any information from these two subsections in their conclusions and therefore I think the meaning of these sections is lost. Either these sections (and the conclusions) should be expanded to include the logic and comparison of the relevant figures, or could be omitted from the paper completely without losing impact (as they are currently difficult to understand - at least by me). You already mention that an additional paper is being prepared about the dribble method, so perhaps that would be the place for a more expansive discussion.

**Author response:** Sections 4.3.2 and 4.3.3 are indeed the most non-straightforward parts of the manuscript.

We imagine colleagues in the CESM and E3SM communities whose work has been influenced by the paper of Williamson (2013) – "The effect of time steps and time‐scales on parametrization suites" (DOI: 10.1002/qj.1992) – would find the results in section 4.3.3 intriguing, while researchers who do not work with these two models might not feel a strong motivation to investigate the impact of the convection scheme's built-in time scale "tau". Having considered comments on Section 4.3.3 from both referees, we choose to move most of the contents in that section to the appendix, leaving only a brief discussion pointing out that the ratio of $\Delta t/\tau$ alone cannot explain the time step

sensitivity of convective activities in our model.

We prefer to leave the short discussion on the "dribbling" experiment in the revised manuscript because comparing v1_Dribble with the simulations v1_CPL+DeepCu_Shorter and v1_CTRL discussed earlier in the manuscript reveals the previously unknown results that the frequency of coupling between the stratiform cloud subcycles and the rest of EAMv1 is the primary reason for cloud-related sensitivities in the subtropical marine stratocumulus regions while the step sizes used for deep convection and/or its coupling with other processes have significant impact in the trade cumulus regions and along the equator (shown by Figure 13 in the original manuscript). These helps to answer the "what step sizes caused what changes" question and hence address the attribution theme of this study, although the question "why does the model behave this way" still needs to be answered by follow-up papers. Nevertheless, to address Dr. Barrett's comment, we have made revisions in Section 4.3.2 and the related figures and captions to better guide the readers through our reasoning.

**Referee comment:** 5) The authors state that the cause of the time step sensitivity coming from outside, rather than within, the subcycled physics is "counter-intuitive" (abstract and line 279). However, this is exactly the mechanism for time step sensitivity that I reported in Barrett et al. (2019) from cloud-resolving model simulations (with time step ranges from 1-15 seconds). I agree that it was initially counter intuitive, but is related to the requirement for the cloud parameterizations to react to a large push away from equilibrium by condensation (resulting from cooling imposed by the dynamics and radiation tendencies). I therefore suggest:

5.1) Adding a sentence that this is potentially caused by condensation/cloud formation in the model and the sensitivity does not get removed even with time steps of only a few seconds (which are not achievable in climate simulations), strengthening the argument for the importance of finding a good numerical solution to this problem (perhaps dribbling...)

**Author response:** We removed the word "counter-intuitive" in the abstract and at line 279 of the original manuscript, as whether a model developer or user would find those results counter-intuitive would probably depend strongly on their prior research experience.

Regarding the observed strong sensitivity of simulated clouds to step sizes outside the subcycles, our recent follow-up work has suggested that the model sensitivities over the mid-latitude storm tracks indeed are closely related to the strong condensation the referee pointed out, while the sensitivities in the subtropical marine stratocumulus regions are more closely related to the positive feedback between cloud liquid amount and cloud-top radiative cooling. We will describe these in detail in the next manuscript.

**Referee comment:** 5.2) Including information in the model description about which part of the model diagnoses the cloud fraction and condensation (i.e. whether this occurs inside or outside of the microphysics+macrophysics and how it is treated within the subcycling setups)

**Author response:** A clarification is added to the model description part (Section 2.1) that CLUBB diagnoses cloud fraction and effectively does the large-scale condensation calculation using the predicted sub-grid probability distribution of heat, water, and vertical velocity, meaning that the condensation and cloud fraction calculations are done at intervals of $\Delta t_{\mathrm{macmic}}$= 5 min.

**Referee comment:**
Typos found:
Line 90: this *is* tied to
Line 440: The mechanisms *under* → *behind*

**Author response:** Both have been corrected. Thanks for pointing them out.
* * *

---

## Author Comment (AC2) · 18 Jan 2021

**Referee comment:** In this manuscript, the authors investigate the sensitivity to model timestep of the mean climate within the EAMv1 model. They then methodically scrutinize the response to variations of the model sub-component timesteps in a series of experiments that are designed to attribute the root cause of the model sensitivity to the individual components.

Perhaps unsurprisingly to those familiar with model development, the largest deviations can be attributed to the parametrizations of clouds and moist convection. Perhaps less predictable is how and where these deviations are - in part due to the direct change

to the sub-timestepping of the microphysics scheme and in part arising from timestep sensitivities of their coupling to other components.

The investigations performed are specific to the EAMv1 atmospheric system and in particular to the specific physics parametrizations, dynamical core and transport formulations and split-explicit temporal coupling. That said, the methodology would translate to other modelling systems and formulations, potentially providing a useful framework for identifying physics or dynamics components that are not well resolved or are poorly formulated for a chosen model timestep length. The method can also help in identifying poorly performing components of the system which might otherwise be hidden or explained away with model tuning. For those without an interest in the specific behaviour of EAM, the most valuable take away message is that developing and employing an approach such as this will help to understand deficiencies and biases when developing weather and climate models.

The manuscript is very well laid out, with a clearly constructed story (with an exception noted in comment 5). I would recommend this article for publication subject to consideration of the minor comments below.

**Author response:** We greatly appreciate the referee's positive assessment of the manuscript, especially their recognition of the general value of our work to the weather and climate modeling community. Our detailed responses to the referee's questions and suggestions are provided below.

**Referee comment:** General comments
1) On L80, the authors mention the passing of tendencies between different components of the atmospheric model and provide the example of the physics tendencies being passed to the resolved dynamics. It is implied that there are other such instances. Although these aren't the subject of the investigation, it would be useful to know what the other instances are, or if these are too numerous/complex to list, then to clarify if there are cases where different physics components pass their tendencies to

subsequent physics components. In particular, do those physics components (clouds, microphysics, convection) which are the focus of the sensitivity studies have such a dependency? If they do, are there any consequences to this that should be borne in mind when interpreting the conclusions about timestep choices between the dependent schemes. If they don't, it would be useful to have that made explicit.

**Author response:** The following clarification and discussion are added to the model description part (Section 2.1) of the revised manuscript:

In terms of the coupling among the coarse-grained components shown in Figure 1a of the original manuscript, the authors are aware of three instances in which tendencies are passed along. These are:

- For the coupling between the parameterized physics and the resolved dynamics, tendencies of temperature and momentum caused by the entire parameterization package are provided to the dynamical core together with the "before-physics" atmospheric state. These are used to update the state variables before each vertical remapping step $\Delta t_{\text{remap}}$. This method of physics-dynamics coupling is depicted in Figure 2b of Zhang et al. (2018, DOI:10.5194/gmd-11-1971-2018) and also discussed in Lauritzen and Williamson (2019, DOI:10.1029/2018MS001549).

- Sensible heat fluxes and moisture fluxes at the Earth's surface are calculated in the "Misc. processes" box in Figure 1a of the original manuscript. The fluxes are not immediately applied to update the atmospheric state; rather, they are passed into the stratiform cloud macro/microphysics subcycles and used as boundary conditions for CLUBB.

- Deep convection is assumed to detrain a certain amount of cloud liquid, causing a source of stratiform cloud condensate. The detrainment-induced tendency of stratiform cloud liquid mass concentration is not applied within or immediately

after the deep convection parameterization but passed into the stratiform cloud macro/microphysics subcycles. After CLUBB has operated, detrainment-induced cloud mass tendency is partitioned into liquid and ice phases using the current temperature values; temperature tendency corresponding to the effective phase change is diagnosed; cloud droplet and crystal number tendencies are derived from the partitioned mass tendencies using assumed cloud particle sizes. These tendencies of cloud liquid and ice as well as temperature are used to update the model state variables before the state variables are provided to the aerosol activation and cloud microphysics parameterization.

In this study we did not touch these parts of the model code. All three cases described above involve passing tendencies of some processes (that are discretized with longer step sizes) to subsequent processes that are subcycled (i.e., use shorter step sizes). The spirit of this method resembles the "sequential splitting" advocated in Beljaars et al. (2004) and Beljaars et al. (2018) as well as the "sequential-tendency splitting" defined in Donahue and Caldwell (2018). The method leads to a tighter coupling as the subcycled processes "feel" the influence of the preceding processes and respond at the shorter intervals; this tighter coupling was exactly our motivation for the "v1_Dribble" simulation described in Section 4.3.2. On the other hand, the processes that are sources of the tendencies only respond to the subcycled processes at longer intervals; the temporal truncation error associated with these longer time steps can have a rather direct impact on the subcycled processes through those tendencies and trigger responses in the subcycled processes.

**Referee comment:** 2) Figure 1: This is a useful schematic to quickly capture the timestepping process. I'm not sure I can see from this where/when the state is updated though. Is it at every point where State is labelled? Could this be made a bit clearer?

**Author response:** Thanks for asking about this. We have revised all schematics and their captions in the manuscript to explicitly depict where/when the model state is

updated.

**Referee comment:** 3) On first inspection it is quite surprising how little an impact there is in the $\Delta t_{\mathrm{macmic}}$ that can be directly attributed to the reduction of $\Delta t_{\mathrm{macmic}}$. However, it is noted in the model overview (L74) that the microphysics includes dynamic substepping and so the timestep used within the microphysics itself may already be shorter than 5 minutes - particularly where numerical stability requires it. While this doesn't negate the conclusions, it might be worth re-stating this feature of the microphysics when discussing this lack of sensitivity. Do the authors have any information regarding the minimum timestep that is used or the number of hydrometeor sedimentation substeps that might add to that understanding?

**Author response:** A clarification is added to the revised manuscript that the dynamic substepping mentioned at L74 of the original manuscript is used only for the sedimentation of hydrometeors; most of the processes in the microphysics parameterization, including for example autoconversion, accretion, and self-collection etc., are calculated using the forward Euler method method with a fixed step size of $\Delta t_{\mathrm{macmic}}$ and parallel splitting. Some further details can be found in Section 2 of Santos et al. (2020, DOI: 10.1029/2019MS001972).

So far we have not done much analysis on the behavior of cloud microphysics in our simulations, and hence unfortunately do not have concrete numbers to help answer the referee's question about minimum step sizes and number of hydrometeor sedimentation substeps. The adaptive sedimentation time steps might be a cause of the lack of sensitivity to $\Delta t_{\mathrm{macmic}}$ in the subtropical low clouds. Another factor to consider is the metrics (physical quantities and their statistics) that are used to assess the time step sensitivity. While the cloud fraction and radiative effects associated with subtropical low clouds generally appear to be much less sensitive to $\Delta t_{\mathrm{macmic}}$ than to step sizes outside the subcycles (Figures 7 and 8 in the original manuscript), the zonal mean specific humidity shown in Figure 6 of the original manuscript does appear to be more sensitive. A somewhat similar example can be found in Section 7 and Figures 13–15

in Santos et al. (2020), where the microphysics time steps were shortened to 1 s; only very minor changes were found in 3-year mean geographical distribution of precipitation, surface temperature, and cloud radiative effects while the differences in the zonal mean vertical distribution of rain mass are substantially larger. (A caveat to mention here is that specific details of the E3SMv1 results from Santos et al. (2020) cannot be directly compared to our results here as our v1_MacMic_Shorter simulation also reduced time steps of CLUBB and its coupling to cloud microphysics.) Some comments are added to the revised manuscript.

**Referee comment:** 4) Regarding the sensitivity of the tropical upper troposphere to the cloud macro and microphysics, there are a number of ways in which the cloud microphysics can directly and indirectly influence this region. See for example figure 1 of Hardiman, Steven C., et al. "Processes controlling tropical tropopause temperature and stratospheric water vapor in climate models." Journal of Climate 28.16 (2015): 6516-6535.

**Author response:** Thanks for pointing this out. A brief comment and some references have been added to the revised manuscript.

**Referee comment:** 5) Section 4.3.3 - Deep convection: This section is at a slight tangent to the rest of the paper and as a result it took me several reads through to be able to absorb the conclusions. Unlike the preceding sections that look to attribute sensitivities, this section attempts to investigate the reason behind the sensitivity of the deep convection to timestep and timescale and then contrasts the findings with the arguments of Williamson (2013). This feels like a half-hearted attempt, with the authors themselves acknowledging more work is needed to fully understand this. I am tempted to suggest removal of this section, however, once I had digested it I did find it interesting and so I would suggest that this goes in an appendix. If the authors do decide to retain the section (in its current location or elsewhere), then please could they clearly signpost at the start of this section that it is a change in direction from the previous stated aims of the paper and is looking beyond attribution?

**Author response:** Following the referee's suggestion, we have moved Figure 14 of the original manuscript and the discussions of the figure to an appendix.

Regarding the referee's point that Section 4.3.3 goes beyond the scope of attribution, our original thought was that the discussion on deep convection had an attribution component because for sensitivities in convective activities, there was the question whether they were caused by the step size of the convection parameterization itself or the step size with which deep convection was coupled with other processes. The discussion in Section 4.3.3 was intended to make an attempt to answer that question. But after pondering on the referee's comment and thinking further about our interpretation of the paper by Williamson (2013), we do agree that the referee has a good point. We have revised the section to clean up the wording and logic.

**Referee comment:**
Typos
L147: 'Working' not 'Wording'
L268: 'Tropical' not 'Topical'

**Author response:** Thanks. These have been corrected.

---

## Author Response (AR1)

RE: Revised manuscript gmd-2020-330

Dear Dr. Ullrich,

On behalf of the co-authors, I hereby submit a revised version of the manuscript entitled "Quantifying and attributing time step sensitivities in present-day climate simulations conducted with EAMv1" (MS No.: gmd-2020-330).

We would like to thank Dr. Andrew Barrett and the anonymous referee for their appreciation of our work and for their very insightful reviews. Their comments, questions, and suggestions have helped to significantly improve the clarity of the paper and the accuracy of our statements.

The key changes we made to the manuscript included:

- 1. explicitly mentioning in the abstract and conclusions section some of the quantitative results;
- 2. adding clarifications about process coupling in EAMv1, including a summary of the tendency-involved coupling methods, in Section 2.1;
- 3. reworking all schematics to clearly distinguish tendency calculations and state updates during EAMv1's time integration loop;
- 4. moving schematics that were originally in the appendix to the corresponding sections in the main text;
- 5. shortening the section on dynamics and radiation step sizes (4.3.1) and revising the section on dribbling (4.3.2) to help make the discussions easier to follow;
- 6. moving the discussion on the  $\Delta t/\tau$  ratio of deep convection to the appendix; shortening and rewriting section 4.3.3 to present only a brief description of the model sensitivities in the deep convection regime plus some initial thoughts on truncation error in a single process versus error in coupling;
- 7. adding a paragraph in the conclusions section to discuss a recent manuscript by Santos et al. (2020, DOI:10.1002/essoar.10504538.1);
- 8. miscellaneous typo corrections and minor edits.

Our detailed responses to the reviews are included on the following pages. A marked-up version of the revised manuscript is also attached. We hope you find the responses and revision satisfactory.

Thanks a lot for handling the review process for this manuscript. We eagerly await your positive response.

Best regards,

Hui Wan

**Referee #1's Comments and the Author's Responses and Changes in Manuscript**

**Referee comment:** This paper addresses the important and until-now under studied impact of changing time-stepping and physics-dynamics coupling methodologies within numerical models of the atmosphere. In this work, the Energy Exascale Earth System Model is used to investigate the impact of a factor of 6 short-ening in the time step for 10-year duration simulation of the present (year 2000) climate. The shortening of the time step reduces errors associated with truncation errors in time and also allows a tighter coupling of the physical processes which are parameterized separately within the model (such as dynamics, clouds, convection, and radiation). These changes have a small to moderate impact on the overall radiative balance of the model resulting in a warming of the lower troposphere (+0.5K), cooling of the upper troposphere/lower stratosphere (- 0.5K), a decrease in low level relative humidity (1%, locally up to 10%) and a reduction of low level clouds, particluarly in subtropical stratocumulus & shallow-cumulus regions. These changes were determined to result from local thermodynamics changes rather than changed to the circulation.

The authors then further investigate which part of the coupling is responsible for these changes through a series of experiments where only a subset of the model's parameterizations use a shortened timestep. The conclusion of these experiments is that differences come from the calculation of the cloud marco-physics+microphysics, but are not caused by sensitivities within the cloud microphysics+microphysics calcula- tions; rather they are caused by changes to the input values to the cloud macro- physics+microphysics parameterization, which result from changing time step length in other parts of the model physics (e.g. larger temperature increments due to longer radiation or dynamics time steps). In an attempt to study this effect, they briefly introduce the "dribble" method of substepping the cloud microphysics with a shorter time step and providing partial updates of the model prognostic variables to each substep based on their change since the last time step.

I find the study to be well constructed, the paper well written and the figures generally clear and justifying inclusion in the paper. I find the results convincing and have no significant concerns regarding the method nor the scientific conclusions of the paper. However, I have a few suggestions for making the paper clearer and improving the clarity of the messages within it. In my opinion the paper is already worthy of publication and my suggestions are intended to improve the clarity of some parts of the paper.

**Author's response:** We greatly appreciate Dr. Barrett's very positive assessment and insightful review. Our responses to his suggestions are detailed below.

**Referee comment:** 1) I suggest including quantification of the changes to temperature, RH and clouds in both the abstract and conclusions so that the scale of the time step sensitivity is easily findable.

Author's response: The suggested changes are included in the revised manuscript.

Author's changes in manuscript: Please see page 1 (lines 4–9), pages 30 (last line) and page 32 (first paragraph) of the marked-up version of the revised manuscript.

Referee comment: 2) I do not see the value of including information about EAMv0 model output in the

paper. The authors should think about whether it is really important to include the description of this model version and the changes in biases between v0 and v1. I didn't find that this information added (or detracted) from my understanding of the time step sensitivity.

**Author's response:** We agree the v0 results do not provide much help in understanding time step sensitivity in v1, as the parameterizations and the vertical resolution in the two model versions are both substantially different. We chose to, and still would like to, keep the v0 results in the manuscript because this little part is intended for a slightly different audience, namely colleagues who have been developing, tuning, and evaluating EAM with a focus on its fidelity in reproducing the observed climate but who might have not thought much about time step sensitivity before reading our manuscript. The v0 results presented here are intended to provide these colleagues with a quantitative assessment of the relative magnitude of the time step sensitivity compared with the changes in model biases when EAM was evolved from v0 to v1.

Author's changes in manuscript: We respectfully chose to keep the v0 results in the manuscript.

**Referee comment:** 3) I found the schematic figure 2 incredibly valuable - showing which sets of simulations I should compare to see what effect. However, the text-based descriptions of the different time-stepping (coupling) strategies I found confusing and made my own notes/sketches when reading the paper. Only at the end of reading the paper did I find the schematics in figures A1 & A2. In this case the figures are much easier to compare than long sections of text and I would therefore suggest adding more explicit references to the explanatory figures at the beginning of each block of text, rather than at the end. (a specific example: more the reference to Figure A2b from line 322 to line 314 (perhaps moving the whole sentence)).

Author's response: We are glad to hear Dr. Barrett's appreciation of the value of the schematics, and we agree that these are more straightforward to read than the text.

Author's changes in manuscript: Following Dr. Barrett's suggestion and example, we have revised the manuscript and moved references to the schematics to the beginning of each block of text description, and added references to the schematics in the various figure captions. We also moved schematics that were originally in the Appendix to the main body of the revised manuscript so that they are placed closer to the text descriptions of the corresponding simulations.

Author's response: Sections 4.3.2 and 4.3.3 are indeed the most non-straightforward parts of the manuscript.

**Referee comment:** 4) The sections 4.3.2 (dribble method) and 4.3.3 (shorter tau in convective param.) are confusing to me. The method is introduced, the results are very briefly summarized and a summary sentence is reached. However, I do not follow which figures should be compared to reach the same conclusions as the authors. Furthermore, the authors do not include any information from these two subsections in their conclusions and therefore I think the meaning of these sections is lost. Either these sections (and the conclusions) should be expanded to include the logic and comparison of the relevant figures, or could be omitted from the paper completely without losing impact (as they are currently difficult to understand - at least by me). You already mention that an additional paper is being prepared about the dribble method, so perhaps that would be the place for a more expansive discussion.

We imagine colleagues in the CESM and E3SM communities whose work has been influenced by the paper of Williamson (2013, "The effect of time steps and time-scales on parametrization suites", DOI: 10.1002/qj.1992) would find the results in section 4.3.3 intriguing, while researchers who do not work with these two models might not feel a strong motivation to investigate the impact of the convection scheme's built-in time scale "tau". Having considered comments on Section 4.3.3 from both referees, we chose to shorten and rewrite section 4.3.3 to present only a brief description of the model sensitivities in the deep convection regime plus some initial thoughts on truncation error in a single process versus error in coupling.

We prefer to leave the short discussion on the "dribbling" experiment in the revised manuscript because comparing v1\_Dribble with v1\_CTRL reveals the previously unknown results that the frequency of coupling between the stratiform cloud subcycles and the rest of EAMv1 is the primary reason for cloud-related sensitivities in the subtropical marine stratocumulus regions (shown by the left column of Figure 13 in the original manuscript and the left column of Figure 16 in the revised version). This helps to answer the "what step sizes caused what changes" question and hence address the attribution theme of this study, although the question "why does the model behave this way" still needs to be answered by follow-up papers. Nevertheless, to address Dr. Barrett's comment, we have made revisions in Sections 4.3.2 and 4.3.3, as well as the related figures and captions, to better guide the readers through our reasoning.

**Author's changes in manuscript:** We have significantly revised Sections 4.3.2 and 4.3.3. Please see pages 26–30, lines 409–480, of the marked-up version.

**Referee comment:** 5) The authors state that the cause of the time step sensitivity coming from outside, rather than within, the subcycled physics is "counter-intuitive" (abstract and line 279). However, this is exactly the mechanism for time step sensitivity that I reported in Barrett et al. (2019) from cloud-resolving model simulations (with time step ranges from 1-15 seconds). I agree that it was initially counter intuitive, but is related to the requirement for the cloud parameterizations to react to a large push away from equilibrium by condensation (resulting from cooling imposed by the dynamics and radiation tendencies). I therefore suggest:

5.1) Adding a sentence that this is potentially caused by condensation/cloud formation in the model and the sensitivity does not get removed even with time steps of only a few seconds (which are not achievable in climate simulations), strengthening the argument for the importance of finding a good numerical solution to this problem (perhaps dribbling...)

**Author's response:** We removed the word "counter-intuitive" in the abstract and at line 279 of the original manuscript, as whether a model developer or user would find those results counter-intuitive would probably depend strongly on their prior research experience.

Regarding the observed strong sensitivity of simulated clouds to step sizes outside the subcycles, our recent follow-up work has suggested that the model sensitivities over the mid-latitude storm tracks indeed are closely related to the strong condensation the referee pointed out, while the sensitivities in the subtropical marine stratocumulus regions are more closely related to the positive feedback between cloud liquid amount and cloud-top radiative cooling. We will describe these in detail in the next manuscript.

Author's changes in manuscript: We removed the word "counter-intuitive" from the abstract (line 14 of

the marked-up version) and removed "surprisingly" from the conclusions section (page 32, line 512).

**Referee comment:** 5.2) Including information in the model description about which part of the model diagnoses the cloud fraction and condensation (i.e. whether this occurs inside or outside of the micro-physics+macrophysics and how it is treated within the subcycling setups)

**Author's response and changes in manuscript:** We added the following clarification to Section 2.1, please see page 6, lines 122-124, of the marked-up version:

"CLUBB diagnoses cloud fraction and effectively does the large-scale condensation calculation using its predicted sub-grid probability distribution functions of heat, water, and vertical velocity. This means the condensation and cloud fraction calculations are done at intervals of  $\Delta t_{\text{macmic}} = 5$  min."

**Referee comment:** Typos found: Line 90: this \*is\* tied to Line 440: The mechanisms \*under\* → \*behind\*

Author's response and changes in manuscript: Both have been corrected. Thanks for pointing them out.

\_\_\_\_\_

**Referee #2's Comments and the Author's Responses and Changes in Manuscript**

**Referee comment:** In this manuscript, the authors investigate the sensitivity to model timestep of the mean climate within the EAMv1 model. They then methodically scrutinize the response to variations of the model sub-component timesteps in a series of experiments that are designed to attribute the root cause of the model sensitivity to the individual components.

Perhaps unsurprisingly to those familiar with model development, the largest deviations can be attributed to the parametrizations of clouds and moist convection. Perhaps less predictable is how and where these deviations are - in part due to the direct change to the sub-timestepping of the microphysics scheme and in part arising from timestep sensitivities of their coupling to other components.

The investigations performed are specific to the EAMv1 atmospheric system and in particular to the specific physics parametrizations, dynamical core and transport formulations and split-explicit temporal coupling. That said, the methodology would translate to other modelling systems and formulations, potentially providing a useful framework for identifying physics or dynamics components that are not well resolved or are poorly formulated for a chosen model timestep length. The method can also help in identifying poorly performing components of the system which might otherwise be hidden or explained away with model tuning. For those without an interest in the specific behaviour of EAM, the most valuable take away message is that developing and employing an approach such as this will help to understand deficiencies and biases when

developing weather and climate models.

The manuscript is very well laid out, with a clearly constructed story (with an exception noted in comment 5). I would recommend this article for publication subject to consideration of the minor comments below.

**Author's response:** We greatly appreciate the referee's positive assessment of the manuscript, especially their recognition of the general value of our work to the weather and climate modeling community. Our detailed responses to the referee's questions and suggestions are provided below.

\_\_\_\_\_

**Referee comment:** General comments**

1) On L80, the authors mention the passing of tendencies between different components of the atmospheric model and provide the example of the physics tendencies being passed to the resolved dynamics. It is implied that there are other such instances. Although these aren't the subject of the investigation, it would be useful to know what the other instances are, or if these are too numerous/complex to list, then to clarify if there are cases where different physics components pass their tendencies to subsequent physics components. In particular, do those physics components (clouds, microphysics, convection) which are the focus of the sensitivity studies have such a dependency? If they do, are there any consequences to this that should be borne in mind when interpreting the conclusions about timestep choices between the dependent schemes. If they don't, it would be useful to have that made explicit.

**Author's response and changes in manuscript** The following clarification and discussion have been added to the model description part (Section 2.1) of the revised manuscript. Please see pages 7–8, lines 148–172 of the marked up version.

In terms of the coupling among the coarse-grained components shown in Figure 1, we are currently aware of three instances in which the model state and its tendencies are both passed to subsequently calculated components. These instances are:

- For the coupling between the parameterized physics and the resolved dynamics, tendencies of temperature and momentum caused by the entire parameterization suite are provided to the dynamical core. These are used to update the state variables before each vertical remapping step Δtremap. This method of physics-dynamics coupling is depicted in Figure 2b of Zhang et al. (2018) and also discussed in Lauritzen and Williamson (2019).
- Sensible heat fluxes and moisture fluxes at the Earth's surface are calculated in the "Misc. processes" box in Figure 1. The fluxes are not immediately applied to update the atmospheric state; rather, they are passed into the stratiform cloud macro/microphysics subcycles and used as boundary conditions for CLUBB.
- Deep convection is assumed to detrain a certain amount of cloud liquid, causing a source of stratiform cloud condensate. The detrainment-induced tendency of stratiform cloud liquid mass concentration is not applied within or immediately after the deep convection parameterization but passed into the stratiform cloud macro/microphysics subcycles. After CLUBB has operated, detrainment-induced cloud mass tendency is partitioned into liquid and ice phases using the current temperature values; temperature tendency corresponding to the effective phase change is diagnosed; cloud droplet and

crystal number tendencies are derived from the partitioned mass tendencies using assumed cloud particle sizes. These tendencies of cloud liquid and ice as well as temperature are used to update the model state variables before the state variables are provided to the aerosol activation and cloud microphysics parameterization.

All three cases described above involve passing tendencies of some processes (that are calculated with longer step sizes) to subsequent processes that are subcycled (i.e., use shorter step sizes). The spirit of this method resembles the "sequential splitting" method advocated in Beljaars et al. (2004) and Beljaars et al. (2018) as well as the "sequential-tendency splitting" method defined in Donahue and Caldwell (2018). The method leads to a tighter coupling as the subcycled processes "feel" the influence of the preceding processes and respond at the shorter intervals; this tighter coupling is the motivation for the "v1\_Dribble" simulation described in Section 4.3.2. On the other hand, the processes causing the tendencies respond to the subcycled processes only at longer intervals; the temporal truncation errors associated with these longer time steps can be manifested in those tendencies and hence trigger responses in the subcycled processes.

**Referee comment:** 2) Figure 1: This is a useful schematic to quickly capture the timestepping process. I'm not sure I can see from this where/when the state is updated though. Is it at every point where State is labelled? Could this be made a bit clearer?

**Author's response and changes in manuscript:** Thanks for asking about this. We have revised all schematics in the manuscript to explicitly depict where/when the model state is updated. The figure on the next page shows an example of how the old (left panel) and new (right panel) schematics look like. We also added a clarification in Section 2.1 (see page 4, lines 97–99 of the marked-up version):

"(In Figure 2 and additional schematics presented later in the paper, tendency calculations in the physics part are depicted by rectangular boxes with sharp corners while the update of model state is shown by oval shapes.)"

**Referee comment:** 3) On first inspection it is quite surprising how little an impact there is in the  $\Delta t_{\text{macmic}}$  that can be directly attributed to the reduction of  $\Delta t_{\text{macmic}}$ . However, it is noted in the model overview (L74) that the microphysics includes dynamic substepping and so the timestep used within the microphysics itself may already be shorter than 5 minutes - particularly where numerical stability requires it. While this doesn't negate the conclusions, it might be worth re-stating this feature of the microphysics when discussing this lack of sensitivity. Do the authors have any information regarding the minimum timestep that is used or the number of hydrometeor sedimentation substeps that might add to that understanding?

Author's response and changes in manuscript: The following clarification has been added to the revised manuscript (see page 6, lines 125–128 of the marked-up version):

"Within the cloud microphysics parameterization, the sedimentation of hydrometeors uses adaptive substepping but the other processes, including for example autoconversion, accretion, and self-collection of rain drops etc., are calculated using the forward Euler method method with a fixed step size of  $\Delta t_{\text{macmic}}$ . Further details about time stepping in the cloud microphysics parameterization can be found in Section 2 of Santos